# Enhancing Molecular Conformer Generation via Fragment-Augmented Diffusion Pretraining

**Xiaozhuang Song**                                                  *xiaozhuangsong1@link.cuhk.edu.cn*
*School of Data Science*
*The Chinese University of Hong Kong, Shenzhen*

**Yuzhao Tu**                                                        *yuzhaotu@link.cuhk.edu.cn*
*School of Data Science*
*The Chinese University of Hong Kong, Shenzhen*

**Tianshu Yu**                                                       *yutianshu@cuhk.edu.cn*
*School of Data Science*
*The Chinese University of Hong Kong, Shenzhen*

**Reviewed on OpenReview:** *https://openreview.net/forum?id=t5WzHOniAF*

## Abstract

Recent advances in diffusion-based methods have shown promising results for molecular con-
former generation, yet their performance remains constrained by training data scarcity—
particularly for structurally complex molecules. In this work, we present Fragment-
Augmented Diffusion (FRAGDIFF), a data-centric augmentation strategy that incorporates
chemical fragmentation techniques into the pre-training phase of modern diffusion-based
generative models. Our key innovation lies in decomposing molecules into chemically mean-
ingful fragments that serve as building blocks for systematic data augmentation, enabling
the diffusion model to learn enhanced local geometry while maintaining global molecular
topology. Unlike existing approaches that focus on complex architectural modifications,
FRAGDIFF adopts a data-centric paradigm orthogonal to model design. Comprehensive
benchmarks show FRAGDIFF's superior performance, especially in data-scarce scenarios.
Notably, it achieves 12.2–13.4% performance improvement on molecules $3\times$ beyond training
scale through pretraining on fragments. Overall, we establish a new paradigm integrating
chemical fragmentations with diffusion models, advancing computational chemistry work-
flows. The code is available at `https://github.com/ShawnKS/fragdiff`.

## 1 Introduction

The generation of molecular conformers serves as a foundational task in computational chemistry, molecular
biology, and drug discovery (Axelrod & Gomez-Bombarelli, 2022; Abramson et al., 2024). Recent years
have witnessed a paradigm shift in conformer generation from rule-based methods to deep learning-driven
approaches (Havel, 1998; Riniker & Landrum, 2015; Jin et al., 2018; Simm & Hernandez-Lobato, 2020; Zhu
et al., 2022), particularly diffusion-based generative models (Xu et al., 2022; Wang et al., 2023; Guo et al.,
2024). These methods leverage the principles of iterative denoising processes, achieving excellent perfor-
mance in modeling the complex, high-dimensional energy landscapes of molecular structures (Rappé et al.,
1992; Halgren, 1996; Zhou et al., 2023). However, despite these advancements, the effectiveness of modern
approaches remains constrained by limited training data – a pervasive challenge in real-world molecular
datasets (Kirchmeyer et al., 2022; Rotskoff, 2024). This paper investigates a critical research question: *Can
we more effectively leverage existing data to improve the performance of diffusion-based conformer generation
models, particularly in data-scarce scenarios?*

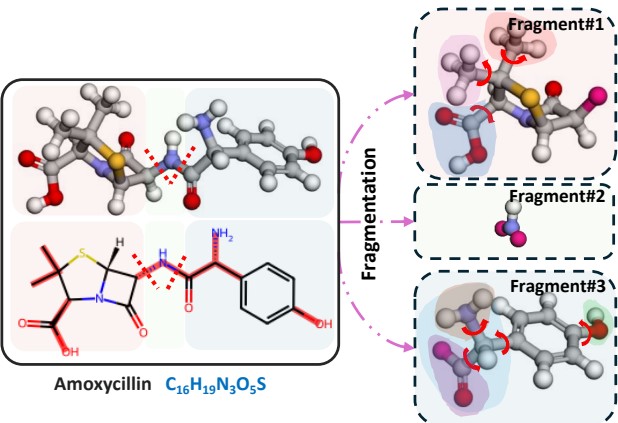

Figure 1: Fragmentation example on *Amoxicillin.*

The inherent compositional and hierarchical structure of molecules provides a natural framework for addressing data scarcity in conformer generation (Kulichenko et al., 2024). Advances in fragment-based drug design have also demonstrated that molecular subsystems often exhibit transferable conformational preferences across different compounds (Brameld et al., 2008). Computational chemistry has long exploited this structure through rule-based fragmentation techniques, which decompose molecules into chemically meaningful components to enable efficient modeling of molecular docking, property prediction, and other tasks (Rappé et al., 1992; Halgren, 1996). For instance, the antibiotic *Amoxicillin* (Fig. 1) can be systematically divided into functional subunits—the $\beta$-*lactam ring*, *thiazolidine ring*, *amino group*, *hydroxyl group*, and *benzene ring*—each governing distinct aspects of molecular conformation and interactions. Despite their interpretability, these traditional approaches lack the flexibility of modern generative models. Conversely, while diffusion-based methods achieve SOTA performance through data-driven learning, they often neglect to encode this foundational chemical prior knowledge (Gordon et al., 2012; Kirchmeyer et al., 2022; Rotskoff, 2024). This represents a potential area for improvement, as molecular fragments' intrinsic geometric constraints could regularize model training and improve sample efficiency, and this gap can be bridged by systematically integrating knowledge-driven fragmentation into both the architecture and training of data-driven models.

Thus, in this work, we present Fragment-Augmented Diffusion (FRAGDIFF), a data-centric augmentation strategy that incorporates traditional chemical fragmentation techniques into the pre-training phase of modern diffusion-based generative models. Our two-phase learning paradigm first pretrains the model on decomposed molecular fragments to capture local fragments' structural patterns, then finetunes on complete molecules to integrate global structural constraints. Comprehensive empirical evaluations of fragmentation pretraining on two distinct diffusion frameworks, GeoDiff and TorDiff, demonstrate consistent improvements across multiple datasets and settings, particularly in data-scarce regimes (Xu et al., 2022; Jing et al., 2022). The success of both paradigms highlights the versatility of fragmentation pretraining in geometric deep learning for molecular conformation generation. Notably, FRAGDIFF also enables effective generation for molecules triple the size of training instances, overcoming conventional scale limitations. This capability demonstrates crucial practical value as real-world drug discovery often requires modeling complex macro-molecular structures. The successful extrapolation suggests our fragment-aware paradigm establishes transferable chemical rules rather than memorizing size-specific patterns, opening new possibilities for generative modeling.

By bridging traditional knowledge-driven methods in computational chemistry with modern generative AI, FRAGDIFF underscores the power of data-centric strategies in improving generative models and highlight the importance of interdisciplinary synthesis in molecule generation research. It serves as a practical tool for practitioners and a conceptual milestone, advocating for deeper integration of domain-specific priors and machine learning in molecular science.

## 2 Related Work

**Diffusion-based Conformer Generation**  Diffusion models have emerged as prominent tools for molecular conformer generation, employing stochastic processes that gradually transform ordered structures into disordered states before learning their reversal through neural networks (Xu et al., 2022; Guo et al., 2024; Hua et al., 2024). While early implementations utilized Cartesian coordinate diffusion with equivariant graph neural networks (Xu et al., 2022; Hoogeboom et al., 2022), these approaches suffer from high computational costs due to iterative denoising of full atomic coordinates (Shi et al., 2021; Luo et al., 2021). Another key advancement comes from Jing et al. (2022), who recognized that molecular flexibility arises primarily from torsional degrees of freedom. Their Torsional Diffusion framework restricts the diffusion process to torsion angles—critical determinants of conformational energy landscapes (Kang et al., 1996)—while preserving bond lengths and angles. This dimensionality reduction strategy enhances sampling efficiency without compromising accuracy, addressing scalability limitations in prior methods.

**Molecular Fragment Decomposition**  Molecular fragment decomposition is a critical concept in computational chemistry, enabling the simplification of complex molecular structures into smaller and more manageable units (Hann et al., 2001; Sliwoski et al., 2014; Sadybekov & Katritch, 2023). This approach facilitates the study of molecular properties and interactions by focusing on individual fragments that retain key chemical characteristics of the parent molecule (Bemis & Murcko, 1996; Jinsong et al., 2024). Fragmentation rules enable systematic molecular decomposition along chemically meaningful boundaries. From a force field perspective, by preserving the local chemical environment around targeted torsions, fragmentation allows for accurate modeling of torsional potentials, ensuring that torsional characteristics can be effectively transferred back to the parent molecule (Horton et al., 2022; D'Amore et al., 2022). Methods like BRICS and RECAP preserve essential functional groups while severing specific bond types (Lewell et al., 1998; Degen et al., 2008), revealing intrinsic properties of molecular subunits and enhancing efficiency by reducing conformational complexity (Liu et al., 2017). By analyzing these fragments, researchers can predict reactivity, optimize drug design, and explore novel chemical spaces with higher precision (Gordon et al., 2012). Beyond conformer generation, fragment-based approaches have been applied to 2D molecular graph generation. Methods like JT-VAE (Jin et al., 2018) and MoLeR (Maziarz et al., 2021) use fragments as building blocks to simplify molecule assembly, differing from our approach that leverages fragments for transfer learning in 3D space. These methods leverage fragmentation primarily to ensure chemical validity, whereas our approach uniquely applies fragment-based knowledge transfer to enhance conformer generation through pre-training. Overall, the integration of molecular fragment decomposition with advanced modeling techniques offers a powerful framework for generating accurate and diverse molecular conformers, ultimately advancing the fields of drug discovery and materials science (Jinsong et al., 2024).

## 3 Methodology

### 3.1 Preliminaries

**Notations and Problem Formulations**  Each molecule with $n$ atoms is represented as an undirected graph $\mathcal{G} = (\mathcal{V}, \mathcal{E})$, where $\mathcal{V} = \{v_i\}_{i=1}^n$ is the set of vertices representing types of atoms, and $\mathcal{E} = \{e_{i,j} \mid (i,j) \subseteq \mathcal{V} \times \mathcal{V}\}$ is the set of edges representing inter-atomic bonds. Each node $v_i \in \mathcal{V}$ contains atomic attributes such as element type and hybridization state. Each edge $e_{i,j} \in \mathcal{E}$ represents a bond between atoms $v_i$ and $v_j$, labeled with its bond type (e.g., single, double, triple). The task of molecular conformer generation involves creating stable and valid 3D conformers $C$ for a given molecular graph $\mathcal{G}$. These conformers are samples from a distribution that reflects the physical and chemical properties related to molecular stability, making it a conditional generative problem. For multiple graphs $\mathcal{G}$, each with its conformers $C$ as independent and identically distributed samples from a Boltzmann distribution, the goal is to learn a generative model $p_\theta(C \mid \mathcal{G})$ that facilitates easy sampling and approximates the Boltzmann function (Hawkins, 2017). Computational approaches diverge in their geometric representations: Methods like GeoDiff (Xu et al., 2022) operate directly in Cartesian coordinate space, while Torsional Diffusion (Jing et al., 2022) employs internal coordinates through torsion angles to model rotational degrees of freedom.

### 3.2 Molecular Fragmentations as Data Augmentation

To enhance molecular representation diversity while preserving key structural and geometric features, we employ fragment-based molecular modeling as a form of data augmentation. By decomposing complete molecular structures into chemically meaningful fragments, we generate augmented data that encode local geometric and chemical information, thereby improving the training of diffusion models. Let the complete molecular structure be represented in a generalized coordinate space $C$, corresponding to internal coordinates such as bond lengths, bond angles, and dihedral angles. Fragmentation decomposes the molecule into $B + 1$ fragments, each associated with a coordinate subspace $\hat{C}_b$ derived from $C$, where $b = 1, \ldots, B + 1$. The fragment coordinates $\hat{C}_b$ retain fragment-specific internal coordinates relevant to their local structures.

Due to data limitations, the true isolated fragment coordinates $C_b$ are often unavailable. Consequently, we approximate them using the corresponding subspaces $\hat{C}_b$ extracted from the complete molecule's coordinates $C$, assuming $\hat{C}_b \approx C_b$. However, fragmentation can modify electronic environments and steric interactions, potentially causing deviations between fragment geometries and those in the complete molecule (Stern et al., 2022; Horton et al., 2022). To reduce these discrepancies, it is crucial to select fragmentation strategies that preserve key chemical features such as conjugation, resonance, steric effects, and hydrogen bonding. By doing so, the fragment properties remain consistent with those of the complete molecule, and $\hat{C}_b$ serves as a reliable proxy for $C_b$. From an information-theoretic perspective, we aim to maximize the mutual information $I(\hat{C}_b; C)$ between the fragment coordinate subspaces $\hat{C}_b$ and the complete molecular coordinate space $C$. Maximizing $I(\hat{C}_b; C)$ ensures that the fragments retain sufficient information about the global structure to accurately reflect molecular properties such as conformeral flexibility and stability. The mutual information is defined as

$$I(\hat{C}_b; C) = H(\hat{C}_b) - H(\hat{C}_b \mid C),$$

where $H(\hat{C}_b)$ is the entropy of the fragment coordinate subspace, and $H(\hat{C}_b \mid C)$ is the conditional entropy given $C$. Different fragmentation methods influence the approximation error and thus affect the mutual information between $\hat{C}_b$ and $C$. Therefore, careful selection of fragmentation strategies is essential to maintain chemical properties and minimize approximation biases. By optimizing these strategies, we can effectively leverage data augmentation in fragment-based molecular modeling to enhance model robustness and accuracy. For a detailed analysis of the impact of different fragmentation methods on these errors, please refer to Appendix B.2.

**Diffusion-based Molecular Conformer Generation**  Diffusion-based models have emerged as powerful tools for molecular conformer generation, particularly in drug discovery (Xu et al., 2022; Guo et al., 2024). These models employ stochastic differential equations (SDEs) to transition molecular structures from their stable equilibrium conformers to highly disordered states through a controlled diffusion process. The reverse process generates samples from the data distribution by iteratively denoising and reconstructing the molecular conformers. In a general framework, diffusion models operate directly on the molecular structures, capturing the geometric properties without being confined to specific internal coordinates such as torsion angles. This approach allows for a comprehensive exploration of the conformeral space, leveraging the intrinsic geometric relationships within the molecule.

**Forward Process**  The forward diffusion process undergoes a transition from their stable equilibrium conformers $C^0$ to a state of increased disorder through a series of diffusion steps, which is modeled as a Markov chain that incrementally adds Gaussian noise to the molecular conformers $C^0$, resulting in a sequence of increasingly disordered states $C^{1:T}$. At each time step $t$, noise is injected according to variance parameters $\beta_t$, which control the scale of perturbation:

$$q(C^{1:T} \mid C^0) = \prod_{t=1}^{T} q(C^t \mid C^{t-1}), q(C^t \mid C^{t-1}) = \mathcal{N}\left(C^t; \sqrt{1 - \beta_t}\, C^{t-1}, \beta_t \mathbf{I}\right).$$

This process ensures a gradual increase in randomness, eventually diffusing the molecular structure into a noise distribution similar to white noise after $T$ iterations.

**Reverse Process**  The reverse diffusion process aims to reconstruct the original molecular conformer $C^0$ from the noisy state $C^T$, guided by a learnable conditional Markov chain. Beginning with chaotic particles $C^T \sim p(C^T)$ drawn from a standard Gaussian distribution, the model iteratively refines the conformers through reverse transitions:

$$p_\theta(C^{0:T-1} \mid \mathcal{G}, C^T) = \prod_{t=1}^{T} p_\theta(C^{t-1} \mid \mathcal{G}, C^t), p_\theta(C^{t-1} \mid \mathcal{G}, C^t) = \mathcal{N}\left(C^{t-1}; \mu_\theta(\mathcal{G}, C^t, t), \sigma_t^2 \mathbf{I}\right),$$

where $\mu_\theta$ is a neural network that estimates the mean of the denoised conformer at each time step $t$, conditioned on the molecular graph $\mathcal{G}$ and the noisy conformer $C^t$. The variance $\sigma_t^2$ may be predefined or learned during training. This reverse process systematically reduces the noise, producing realistic molecular conformers that align with the data distribution. This general diffusion framework enables flexible modeling of molecular structures, capturing both geometric and topological features.

Typically, diffusion frameworks parameterize $\mu_\theta$ to directly predict the denoising direction. Alternatively, the reverse process can be guided by approximating the score function $\nabla_C \log p_t(C)$, which indicates how to move from the current state toward the data distribution. To unify these perspectives, we employ a neural network $\mathbf{s}_\theta(\mathcal{G}, C^t, t)$ to approximate this score function, where $\mathbf{s}_\theta$ is closely related to $\mu_\theta$ through the relationship $\mu_\theta = C^t - \sigma_t^2 \mathbf{s}_\theta$ (Song & Ermon, 2019; Xu et al., 2022; Jing et al., 2022). The score network $\mathbf{s}_\theta$ aims to match the true score function with the loss function defined as:

$$J_{\text{DSM}}(\theta) = \mathbb{E}_t\left[\lambda(t)\mathbb{E}_{C^0 \sim p_0, C^t \sim p_{t|0}(\cdot|C^0)}\left[\left\|\mathbf{s}_\theta(\mathcal{G}, C^t, t) - \nabla_{C^t} \log p_{t|0}(C^t \mid C^0, \mathcal{G})\right\|^2\right]\right].$$

Here, $\lambda(t)$ is a precomputed weighting factor to balance contributions across timesteps. Through this formulation, $\mathbf{s}_\theta(\mathcal{G}, C^t, t)$ explicitly learns to denoise the conformer by aligning with the score function, while maintaining consistency with the mean parameterization $\mu_\theta$. The network thereby recovers stable molecular conformers from noisy initial states.

### 3.3  Roto-translational Equivariant Network

For diffusion models designed for molecular conformer generation, the network architecture typically adopts a roto-translational equivariant design, also known as an SE(3)-equivariant network (Thomas et al., 2018; Geiger & Smidt, 2022; Zhou et al., 2024). This design ensures that the network's outputs are equivariant with respect to rotations and translations, which is an effective approach for modeling molecular structures that are inherently symmetric under such transformations. By incorporating equivariance, the network accurately captures the geometric properties of molecules, leading to more physically plausible and consistent conformer generation.

The SE(3)-equivariant framework operates through distinct strategies: For instance, GeoDiff (Xu et al., 2022) directly models inter-atomic coordinates using equivariant graph neural networks that update node features and positions via message passing, while TorDiff (Jing et al., 2022) focuses on torsional degrees of freedom (dihedral angles within internal coordinates) by applying rotational transformations to substructures around chemical bonds. Both approaches enforce critical symmetries: SE(3)-equivariance ensures invariance to global translations, and spatial inversion symmetry ($p(C) = p(-C)$) preserves physical consistency under mirror reflections. These symmetries are embedded into score functions, where GeoDiff computes SE(3)-invariant scalar updates through edge-wise interactions, and TorDiff encodes pseudoscalar transformations for torsional components. The architectures further ensure that gradient updates to atomic coordinates or torsional angles respect the antisymmetric property under spatial inversion ($\nabla_C \log p_t(C \mid \mathcal{G}) = -\nabla_C \log p_t(-C \mid \mathcal{G})$), aligning with fundamental physical laws. A complete technical description of these frameworks, including their mathematical formulations and implementation details, is provided in Appendix A.1.

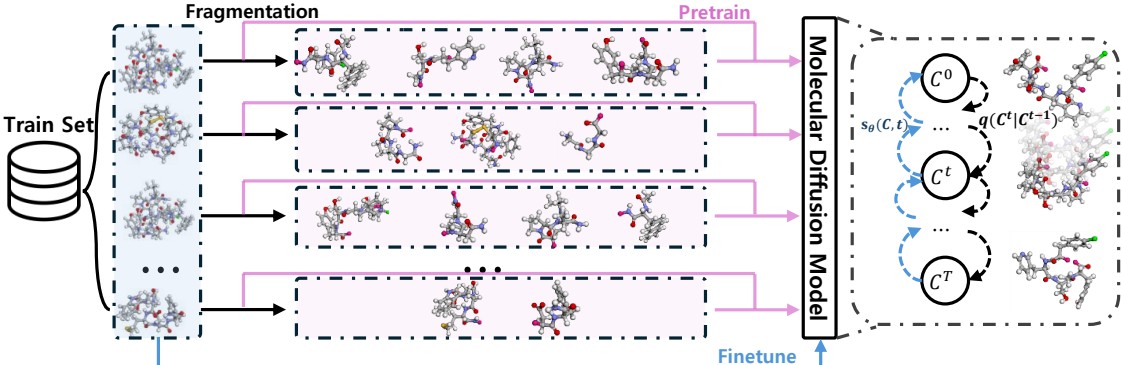

Figure 2: An overview of the FRAGDIFF pipeline. Molecules from the training set will be fragmented based on randomly selected fragmentation edges. The resulting fragments will be further used as augmented data in the pretraining phase.

### 3.4 Fragment-Augmented Diffusion Pretraining

In this work, we introduce a fragment-augmented diffusion pretraining approach, where molecules are decomposed into smaller meaningful fragments using specific rules. Figure 2 illustrates an overview of the fragment-augmented diffusion pretraining pipeline. By pretraining on fragmented molecular data, the model learns rich representations that can be fine-tuned for downstream tasks. This pretraining paradigm aims to enhance the diffusion model's ability by leveraging fragment-level information, leading to better molecular conformer generation.

The fragmentation is guided by identifying key rotatable bonds or functional groups, ensuring that each fragment retains essential chemical and structural information (Jinsong et al., 2024). By focusing on these fragments during pretraining, we enable the model to learn from smaller, more manageable substructures, which can be optimized independently while maintaining global molecular consistency through interactions between fragments. This yields $B + 1$ fragments per molecule (from $B$ decomposition edges), where each fragment constitutes an independent subgraph with preserved local connectivity. Once a molecule is fragmented into multiple fragments, each fragment is treated as an independent subgraph consisting of its own nodes (atoms) and edges (bonds). The conformer generation task for each fragment is performed independently in the pretraining phase. Molecules from the training set are fragmented based on randomly selected fragmentation edges, and the resulting smaller fragments are used as augmented data to pretrain the model.

The loss for each fragment is computed separately, and these losses are averaged to form the total loss function during pretraining. For each molecule $i$ in the dataset containing $N$ samples, we compute fragment-specific losses then aggregate across both fragments and molecules:

$$\mathcal{L}_{\text{pretrain}} = \frac{1}{N} \sum_{i=1}^{N} \frac{1}{B_i + 1} \sum_{b=1}^{B_i+1} \mathbb{E}_t \left[ \left\| \mathbf{s}_\theta(C^{(i)}, t) - \nabla_C \log p_{t|0}(C^{(i)} \mid C^{0,(i)}, \mathcal{G}_b^{(i)}) \Big|_{C=C_b} \right\|^2 \right]$$

where $B_i$ denotes decomposition edges count for molecule $i$, $\mathcal{E}_b^{(i)}$ represents edges in the $b$-th fragment of molecule $i$, $C_b$ presents the coordinates for atoms in fragment $b$ of molecule $i$, $\mathcal{G}_b^{(i)}$ denotes the local graph structure of fragment $b$ in molecule $i$. $\mathbf{s}_\theta(C^{(i)}, t)$ is the neural network parameterized score function that predicts the gradient of the log probability density at time step $t$, and $\nabla_C \log p_{t|0}(C^{(i)})$ represents the true score function of the perturbation kernel, describing the gradient of the log probability density of the noised coordinates given the original coordinates.

### 3.5 Model Training

**Evidence Lower Bound (ELBO) Optimization** Using the probability flow ODE, we compute the likelihood of any sample $C^0$ as $\log p_0(C^0) = \log p_T(C^T) - \frac{1}{2} \int_0^T \frac{d}{dt} \sigma^2(t) \nabla_{C^t} \cdot \mathbf{s}_\theta(\mathcal{G}, C^t, t) \, dt$. (Song & Ermon,

2020; De Bortoli, 2022). Since directly computing the exact log-likelihood $\mathbb{E}[\log p_\theta(C^0|\mathcal{G})]$ is intractable, we resort to maximizing its evidence lower bound (ELBO):

$$\mathbb{E}[\log p_\theta(C^0|\mathcal{G})] = \mathbb{E}\left[\log \frac{p_\theta(C^{0:T}|\mathcal{G})}{q(C^{1:T}|C^0)}\right] \geq -\mathbb{E}_q\left[\sum_{t=1}^T D_{\mathrm{KL}}\big(q(C^{t-1}|C^t, C^0) \,\|\, p_\theta(C^{t-1}|\mathcal{G}, C^t)\big)\right],$$

where $q(C^{t-1}|C^t, C^0)$ is analytically tractable due to the properties of the forward diffusion process. Specifically, for any timestep $t$, the distribution $q(C^t|C^0)$ can be expressed in closed form as:

$$q(C^t|C^0) = \mathcal{N}\left(C^t; \sqrt{\bar{\alpha}_t}\, C^0, (1 - \bar{\alpha}_t)\mathbf{I}\right),$$

where $\alpha_t = 1 - \beta_t$ and $\bar{\alpha}_t = \prod_{s=1}^t \alpha_s$. This formulation allows us to derive $q(C^{t-1}|C^t, C^0)$ analytically and efficiently. Building upon the insights from Ho et al. (2020) and Xu et al. (2022), we can further simplify the ELBO by expressing the KL divergence between Gaussian distributions as a weighted $L_2$ loss between the predicted score function $\mathbf{s}_\theta(\mathcal{G}, C^t, t)$ and the target score $\nabla_{C^t} \log p_{t|0}(C^t \mid C^0, \mathcal{G})$. Consequently, we can independently sample conformers at various timesteps from $q(C^{t-1}|C^t, C^0)$ to optimize the objective more efficiently. By modeling the score function with $\mathbf{s}_\theta$, we enhance the training process and improve the model's ability to generate accurate torsional angles (Xu et al., 2022).

**Pretraining and Finetuning**  The pretraining phase employs fragment-based learning to enhance the diffusion network's ability on generating molecular substructures. Molecules are systematically decomposed into smaller fragments using predefined decomposition rules, and the network is trained on these individual fragments. This fragment-level training enables the network to learn representations of molecular substructures for generating accurate full molecular conformers. Following pretraining, the diffusion network undergoes fine-tuning using data from real-world molecular datasets to accurately handle complete molecular geometries. This holistic fine-tuning process effectively combines fragment-level knowledge with empirical data, optimizing the network's ability to generate accurate and physically plausible molecular conformers.

**Conformer Sampling**  To generate a stable conformer $C^0$ for a given molecular graph $\mathcal{G}$, we begin by sampling a noisy state $C^T$ from the prior distribution $p_T(C^T)$, typically a standard Gaussian distribution. Then, employing the trained reverse process $p_\theta(C^{t-1} \mid \mathcal{G}, C^t)$ as defined earlier, we iteratively sample $C^{t-1}$ from $C^t$ as $t$ decreases from $T$ to 1. At each timestep, the neural network $\mu_\theta(\mathcal{G}, C^t, t)$ predicts the mean of the denoised conformer, guiding the refinement of particle positions. This sequential process gradually transitions the conformer from a chaotic state to an equilibrium state, yielding realistic molecular structures that align with the learned data distribution.

## 4 Experiments

### 4.1 Experimental setup

**Dataset**  We utilize three subsets GEOM-QM9, GEOM-DRUGS, and GEOM-XL from the GEOM dataset (Axelrod & Gomez-Bombarelli, 2022), which provides high-quality conformer ensembles generated using metadynamics in CREST (Pracht et al., 2020). GEOM-QM9 is a dataset featuring significantly smaller molecules with an average of 11 atoms. GEOM-DRUGS represents the most pharmaceutically relevant subset, comprising molecules with an average of 44 atoms. GEOM-XL is created by selecting all species with more than 100 atoms from GEOM-MoleculeNet (Wu et al., 2018), allowing us to evaluate models' generation quality on large molecules. For a detailed statistics for all three datasets are in Appendix D.2.

**Evaluation**  To assess both diversity and quality, we apply two key metrics: Average Minimum RMSD (AMR) and Coverage (COV). These metrics are reported for both Recall (AMR-R, COV-R) and Precision (AMR-P, COV-P). COV-R and AMR-R measure how well the generated ensemble covers the ground-truth ensemble, while COV-P and AMR-P assess the accuracy of the generated conformers. The calculations of COV-R and AMR-R are defined as:

$$\text{AMR-R} := \frac{1}{L} \sum_{l=1}^{L} \min_{k=1}^{K} \text{RMSD}(C_k, C_l^*)$$

$$\text{COV-R} := \frac{1}{L} \sum_{l=1}^{L} \Vdash \min_{k} \text{RMSD}(C_k, C_l^*) < \delta$$

For precision metrics, COV-P and AMR-P are calculated by swapping the roles of the generated and reference sets. These metrics emphasize the quality of the generated conformers. In our evaluations, we set the threshold $\delta$ to 0.5 Å for the GEOM-QM9 dataset and 0.75 Å for the GEOM-DRUGS dataset. Higher COV scores or lower AMR scores indicate more realistic conformers, effectively balancing both diversity and accuracy.

**Baselines** To establish a comprehensive baseline for our proposed data augmentation methods, we compare them against both traditional computational techniques and state-of-the-art deep learning models. Among traditional methods, we utilize RDKit ETKDG (Havel, 1998; Riniker & Landrum, 2015), a widely recognized open-source tool, Metrization (Havel, 1998), a distance geometry-based conformer sampling approach, and OMEGA (Hawkins, 2017), a commercial software known for its continuous development and reliability. In the realm of deep learning, we benchmark against several leading models, including CGCF (Xu et al., 2021), ConfGF (Shi et al., 2021), GeoMol (Ganea et al., 2021), GeoDiff (Xu et al., 2022), Torsional Diffusion (TorDiff) (Jing et al., 2022), and DMCG (Zhu et al., 2022) . We implement our FRAGDIFF approach through two established diffusion frameworks: GeoDiff (Xu et al., 2022) which models molecular geometry in continuous space, and TorDiff (Jing et al., 2022) which specializes in torsional angle sampling. The resulting implementations are denoted as FRAGDIFF-G and FRAGDIFF-T, maintaining the core architectures while integrating our fragment-based pretraining.

**Fragmentation Augmentation Setup** In our main FRAGDIFF implementation, fragmentation edges are identified by combining those recognized by BRICS rules, RECAP rules, and graph-based fragmentation (Algorithm 1) into a unified candidate pool. For a given molecule, we identify all fragmentation-edges and randomly select $B = \min(b, \kappa)$ edges, where $b$ is the total number of fragmentation-edges and $\kappa$ limits the maximum number of selected edges to avoid excessive small fragments. From the resulting $B + 1$ fragments, those with rotatable bonds are used to augment the training set. Our experiments use $\kappa = 5$. During fragmentation, only fragments larger than $z$ atoms are selected for augmentation. This ensures that the resulting fragments retain sufficient structural complexity and chemical information to contribute meaningfully to the training process. To explore the impact of reaction-related bonds on model performance, we also test models generated after removing these bonds, focusing on BRICS and RECAP rules (Lewell et al., 1998; Degen et al., 2008). Detailed introductions of these two chemical rules are provided in the Appendix B.1, and additional results and discussions on how the choice of the minimum fragment size parameter $z$ affects fragmentation statistics are provided in Appendix D.2.

**Experimental environment and Model Setup** The detailed experimental environment and parameter settings for the experiments can be found in Appendix C.

## 4.2 Conformation Generation

**Performance on GEOM-DRUGS** As shown in Table 1, our fragment-augmented models demonstrate substantial improvements over existing approaches. FRAGDIFF-T achieves state-of-the-art performance, attaining the highest mean COV-R (70.07% vs TorDiff's 67.49%) and COV-P (52.87% vs TorDiff's 49.53%), along with the lowest AMR-R (0.609 Å vs 0.634 Å) and AMR-P (0.800 Å vs 0.827 Å). While DMCG outperforms many baselines, FRAGDIFF-T still achieves substantially better results in all metrics, demonstrating the effectiveness of diffusion models with fragmentation pretraining. This consistent superiority across both coverage and accuracy metrics underscores the effectiveness of our fragment-based augmentation strategy. The advantages of our approach are particularly evident when comparing to baseline diffusion models. While

Table 1: Quality of generated conformers for the GEOM-DRUGS test set in terms of Coverage (%) and Average Minimum RMSD (Å) with $\delta = 0.75$ Å. The results of CGCF, CONFGF are borrowed from Shi et al. (2021). The results of GeoMol, OMEGA, ETKDG are borrowed from Jing et al. (2022), the results of DMCG are obtained by evaluating the official checkpoint with the same $\delta = 0.75$ Å, and the rest results are obtained by our own experiments.

| Models | COV-R (%) ↑ | | AMR-R (Å) ↓ | | COV-P (%) ↑ | | AMR-P (Å) ↓ | |
|---|---|---|---|---|---|---|---|---|
| | Mean | Median | Mean | Median | Mean | Median | Mean | Median |
| Metrization | 5.712 | 0.00 | 1.388 | 1.329 | 4.932 | 0.000 | 1.541 | 1.339 |
| CGCF | 5.810 | 0.00 | 1.248 | 1.224 | 0.200 | 0.000 | 1.857 | 1.806 |
| ConfGF | 9.150 | 0.50 | 1.162 | 1.159 | 3.060 | 0.250 | 1.721 | 1.686 |
| ETKDG | 38.40 | 28.60 | 1.058 | 1.002 | 40.90 | 30.80 | 0.995 | 0.895 |
| GeoMol | 44.60 | 41.40 | 0.875 | 0.834 | 43.00 | 36.40 | 0.928 | 0.841 |
| OMEGA | 53.40 | 54.60 | 0.841 | 0.762 | 40.50 | 33.30 | 0.946 | 0.854 |
| DMCG | 57.55 | 59.00 | 0.722 | 0.723 | 37.57 | 35.33 | 0.944 | 0.909 |
| GeoDiff | 45.61 | 49.32 | 0.862 | 0.852 | 21.47 | 14.55 | 1.171 | 1.123 |
| FRAGDIFF-G | 51.56 | 52.34 | 0.847 | 0.838 | 26.13 | 20.63 | 1.095 | 1.084 |
| TorDiff | 67.49 | 75.81 | 0.634 | 0.618 | 49.53 | 47.16 | 0.827 | 0.778 |
| FRAGDIFF-T | **70.07** | **78.35** | **0.609** | **0.588** | **52.87** | **54.17** | **0.800** | **0.749** |

Table 2: Quality of generated conformers for the GEOM-QM9 test set in terms of Coverage (%) and Average Minimum RMSD (Å) with $\delta = 0.5$ Å. The results of CGCF, ConfGF are borrowed from Shi et al. (2021). The results of GeoMol, OMEGA, and ETKDG are borrowed from Jing et al. (2022), the results of DMCG are borrowed from (Zhu et al., 2022), and the rest results are obtained by our own experiments.

| Models | COV-R (%) ↑ | | AMR-R (Å) ↓ | | COV-P (%) ↑ | | AMR-P (Å) ↓ | |
|---|---|---|---|---|---|---|---|---|
| | Mean | Median | Mean | Median | Mean | Median | Mean | Median |
| CGCF | 78.0 | 82.4 | 0.421 | 0.390 | 36.5 | 33.6 | 0.662 | 0.643 |
| ConfGF | 88.4 | 94.3 | 0.267 | 0.268 | 46.4 | 43.4 | 0.522 | 0.512 |
| ETKDG | 85.1 | **100.0** | 0.235 | 0.199 | 86.8 | **100.0** | 0.232 | 0.205 |
| OMEGA | 85.5 | **100.0** | 0.177 | **0.126** | 82.9 | **100.0** | 0.224 | **0.186** |
| GeoMol | 91.5 | **100.0** | 0.225 | 0.193 | 86.7 | **100.0** | 0.270 | 0.241 |
| DMCG | **96.3** | **100.0** | 0.206 | 0.200 | 87.2 | 91.0 | 0.287 | 0.292 |
| GeoDiff | 90.1 | 93.4 | 0.209 | 0.198 | 52.8 | 50.3 | 0.445 | 0.427 |
| FragDiff-G | 91.1 | 95.1 | 0.199 | 0.194 | 56.0 | 55.0 | 0.431 | 0.421 |
| TorDiff | 92.8 | **100.0** | 0.178 | 0.147 | **92.7** | **100.0** | **0.221** | 0.195 |
| FragDiff-T | 93.2 | **100.0** | **0.175** | 0.139 | **93.1** | **100.0** | **0.218** | 0.189 |

GeoDiff achieves 45.61% COV-R and 21.47% COV-P, our FRAGDIFF-G variant improves these to 51.56% and 26.13% respectively, while simultaneously reducing AMR metrics by 0.015 Å (R) and 0.076 Å (P). This demonstrates the generalizability of our fragmentation paradigm across different backbone architectures. Notably, FRAGDIFF-T's performance improvement over TorDiff (2.58-3.34% absolute improvement in coverage metrics with 0.025-0.027 Å AMR reduction) highlights the particular synergy between torsional diffusion frameworks and fragment-based pretraining. The results suggest that our augmentation strategy enables more comprehensive sampling of low-energy conformers while maintaining high structural fidelity, addressing the diversity-accuracy trade-off observed in previous methods like ETKDG (38.40% COV-R) and OMEGA (53.40% COV-R but lower precision).

**Performance on GEOM-QM9** Table 2 presents the performance on the GEOM-QM9 test set, which primarily consists of small molecules, making it a suitable benchmark for evaluating the ability of models to generate accurate conformers for relatively simple molecular structures. The results are evaluated with a threshold of $\delta = 0.5$ Å. Our proposed model, FRAGDIFF-T, achieves the highest or competitive performance on all metrics. DMCG achieves strong performance with 96.3% on COV-R, demonstrating the effectiveness of direct coordinate regression for simpler structures. However, our FRAGDIFF-T still maintains superior performance in other metrics. The performance of FRAGDIFF-T on the GEOM-QM9 dataset highlights its effectiveness in capturing the simpler molecular structures. Notably, FRAGDIFF-G, our fragment-augmented

Table 3: Performance on the GEOM-XL dataset.

| Model | AMR-R ↓ | | AMR-P ↓ | |
|---|---|---|---|---|
| | Mean | Median | Mean | Median |
| RDKit | 2.92 | 2.62 | 3.35 | 3.15 |
| GeoMol | 2.47 | 2.39 | 3.30 | 3.15 |
| TorDiff | 2.05 | 1.86 | 2.94 | 2.78 |
| FRAGDIFF-T | **1.80** | **1.61** | **2.60** | **2.44** |

Table 4: Results of Property Prediction task.

| Method | $E$ | $E_{\min}$ | $\Delta\epsilon$ | $\Delta\epsilon_{max}$ |
|---|---|---|---|---|
| RDKit | 0.92 | 0.65 | 0.37 | 0.80 |
| GeoMol | 0.38 | 0.19 | 0.29 | 0.81 |
| GeoDiff | 0.26 | 0.14 | 0.31 | 0.70 |
| FRAGDIFF-G | 0.24 | **0.13** | 0.29 | 0.64 |
| TorDiff | 0.20 | 0.14 | 0.23 | **0.43** |
| FRAGDIFF-T | **0.19** | **0.13** | **0.20** | **0.43** |

variant of GeoDiff, also demonstrates significant improvements over its baseline. It achieves a mean COV-R of 91.1% (vs. 90.1% for GeoDiff) and a median of 95.1% (vs. 93.4%), along with better AMR-R scores of 0.199Å mean (vs. 0.209Å) and 0.194Å median (vs. 0.198Å). Similar improvements are observed in prediction metrics, with FRAGDIFF-G achieving higher COV-P (56.0% vs. 52.8%) and lower AMR-P (0.431Å vs. 0.445Å). These consistent improvements across all metrics demonstrate that our fragment augmentation strategy effectively enhances GeoDiff's conformer generation capability.

**Performance on GEOM-XL**  To assess generalization capabilities beyond standard benchmarks, we specifically evaluate FRAGDIFF-T on the challenging GEOM-XL dataset containing molecules with $\sim 3\times$ more atoms on average than the training GEOM-Drugs set. As shown in Table 3, the fragment-augmented model demonstrates remarkable improvements over existing approaches. Notably, FRAGDIFF-T reduces the mean AMR-R by 12.2% and median AMR-R by 13.4% compared to the base TorDiff model, establishing new state-of-the-art recall performance. The precision metrics show similar gains, with mean AMR-P improving 11.6% and median AMR-P decreasing 12.2%. The significant performance gap underscores the advantages of our approach: the fragmentation pretraining strategy enhances robustness for handling larger molecular systems beyond training domain scales, while local fragment-based learning captures structural patterns that generalize to extended molecular architectures. These results further validate that fragment-level data augmentation alleviates the scale limitations.

**Property Prediction Tasks**  We adopt the property prediction task setup from Xu et al. (2022); Shi et al. (2021), where 30 molecules from the GEOM-DRUGS dataset are used, with 50 samples generated for each molecule. The PSI4 toolkit is employed to compute the energy ($E$) and HOMO-LUMO gap ($\epsilon$) for each conformer, and comparisons are made with the ground truth for average energy ($E$), minimum energy ($E_{\min}$), average gap ($\Delta\epsilon$), and maximum gap ($\Delta\epsilon_{max}$). As shown in Table 4, our fragment augmentation strategy brings consistent improvements to both GeoDiff and TorDiff. Specifically, FRAGDIFF-G enhances the performance of GeoDiff, reducing the average energy error from 0.22 to 0.20 and the HOMO-LUMO gap error from 0.23 to 0.21. Similarly, FRAGDIFF-T improves upon TorDiff across all metrics, achieving the lowest average energy error (0.19) and HOMO-LUMO gap error (0.20) among all methods. These improvements demonstrate that our fragment augmentation strategy effectively enhances the conformer generation quality of existing methods. By incorporating local structural information through fragmentation, both GeoDiff and TorDiff variants achieve more accurate predictions of quantum chemical properties.

**Model Performance Across Different Training Sample Sizes**  Table 5 illustrates the performance of our proposed models FRAGDIFF-T and FRAGDIFF-G compared to the baseline methods TorDiff and GeoDiff, respectively, across varying training sample sizes. FRAGDIFF-T consistently outperforms TorDiff across all metrics and training sample sizes. For 1000 samples, FRAGDIFF-T achieves a COV-R of 49.39%, which is

Table 5: Quality of generated conformers for the GEOM-DRUGS test set with $\delta = 0.75$ Å on varying available training samples **n**. (Mean)

| Models | | FRAGDIFF-G | | | | GeoDiff | | |
|---|---|---|---|---|---|---|---|---|
| Metric | | **COV-R** | **AMR-R** | **COV-P** | **AMR-P** | **COV-R** | **AMR-R** | **COV-P** | **AMR-P** |
| | 1000 | 13.07 | 1.250 | 7.658 | 1.346 | 8.780 | 1.347 | 2.191 | 1.413 |
| n | 5000 | 15.69 | 1.159 | 10.03 | 1.251 | 13.83 | 1.242 | 6.327 | 1.383 |
| | 10000 | 30.30 | 1.051 | 21.02 | 1.166 | 16.78 | 1.175 | 9.196 | 1.261 |
| Models | | FRAGDIFF-T | | | | TorDiff | | |
| Metric | | **COV-R** | **AMR-R** | **COV-P** | **AMR-P** | **COV-R** | **AMR-R** | **COV-P** | **AMR-P** |
| | 1000 | **49.39** | **0.7928** | **33.84** | **1.0455** | 34.60 | 0.9833 | 20.84 | 1.1897 |
| n | 5000 | **51.17** | **0.7519** | **34.51** | **1.0389** | 44.61 | 0.8209 | 25.77 | 1.1104 |
| | 10000 | **62.82** | **0.6736** | **43.10** | **0.9081** | 52.76 | 0.7507 | 33.88 | 1.0458 |

Table 6: Quality of generated conformers for the GEOM-DRUGS test set in terms of Coverage (%) and Average Minimum RMSD (Å) with $\delta = 0.75$ Å with 5000 training samples.

| Models | COV-R (%) ↑ | | AMR-R (Å) ↓ | | COV-P (%) ↑ | | AMR-P (Å) ↓ | |
|---|---|---|---|---|---|---|---|---|
| | **Mean** | **Median** | **Mean** | **Median** | **Mean** | **Median** | **Mean** | **Median** |
| FRAGDIFF-T | **51.17** | **50.10** | **0.7519** | **0.7503** | **34.51** | **21.61** | **1.0389** | **1.0224** |
| w/o BRICS | 50.85 | 49.61 | 0.7568 | 0.7575 | 33.93 | 20.57 | 1.0461 | 1.0313 |
| w/o RECAP | 49.38 | 48.42 | 0.7609 | 0.7639 | 34.18 | 21.21 | 1.0420 | 1.0247 |
| w/o B & R | 48.60 | 46.89 | 0.7684 | 0.7708 | 33.74 | 19.86 | 1.0492 | 1.0377 |

42% higher than TorDiff's 34.60%, and reduces AMR-R to 0.7928 Å compared to TorDiff's 0.9833 Å. When the training size increases to 5000 samples, FRAGDIFF-T maintains its advantage with a COV-R of 51.17%, outperforming TorDiff's 44.61%, and lowering AMR-R to 0.7519 Å versus TorDiff's 0.8209 Å. At the largest sample size of 10000 samples, FRAGDIFF-T achieves a COV-R of 62.82%, which is 19% higher than TorDiff's 52.76%, and further decreases AMR-R to 0.6736 Å compared to TorDiff's 0.7507 Å. Similarly, FRAGDIFF-G demonstrates superior performance over GeoDiff across all metrics. At 1000 samples, FRAGDIFF-G attains a COV-R of 13.07%, which is 48.86% higher than GeoDiff's 8.78%, and reduces AMR-R to 1.250 Å compared to GeoDiff's 1.347 Å. With 5000 samples, FRAGDIFF-G achieves a COV-R of 15.69%, surpassing GeoDiff's 13.83%, and an AMR-R of 1.159 Å versus GeoDiff's 1.242 Å. At 10000 samples, FRAGDIFF-G's COV-R increases to 30.30%, which is 80% higher than GeoDiff's 16.78%, and AMR-R decreases to 1.051 Å compared to GeoDiff's 1.175 Å. Additionally, both FRAGDIFF-T and FRAGDIFF-G consistently deliver superior performance in COV-P and AMR-P metrics across all sample sizes, demonstrating their enhanced capability to generate conformer ensembles that are both diverse and accurate, even with limited data. These results highlight the robustness and scalability of our FRAGDIFF models, particularly in data-scarce environments, while also demonstrating their capacity to improve with larger datasets.

### 4.3 Further Experimental Results

**Impact of Chemical Fragmentation** We explore the individual contributions of different fragmentation strategies by selectively removing edge types from the candidate pool. Here, 'removing' means excluding the bonds identified by that method from consideration as potential cutting points during fragmentation. We further explore the impact of chemical fragmentation strategies on the performance of FRAGDIFF-T. Table 6 shows the effect of removing BRICS and RECAP reaction edges on conformer generation. The full FRAGDIFF-T model, which includes both, achieves the best performance with a mean COV-R of 51.17% and COV-P of 50.10%. Removing BRICS (w/o BRICS) has a larger impact on precision, reducing COV-P to 33.93% and increasing AMR-P to 1.0461 Å, indicating BRICS edges are crucial for precision. In contrast, removing RECAP (w/o RECAP) affects recall more, with COV-R dropping to 49.38% and AMR-R rising to 0.7609 Å, showing RECAP edges are key for coverage. The largest performance drop occurs when both BRICS and RECAP edges are removed (w/o B & R), with COV-R at 48.60% and COV-P at 33.74%, highlighting the complementary roles of BRICS and RECAP-related bonds. These results underscore the potential of incorporating chemical semantic knowledge, such as BRICS and RECAP reaction edges, in enhancing chemical generative models, as both play crucial roles in generating diverse and accurate conformers.

Table 7: Training with and without conformer matching (CM) on the GEOM-DRUGS test set in terms of Coverage (%) and Average Minimum RMSD (Å) with $\delta = 0.75$ Å.

| Samples | Models | COV-R (%) ↑ | | AMR-R (Å) ↓ | | COV-P (%) ↑ | | AMR-P (Å) ↓ | |
|---|---|---|---|---|---|---|---|---|---|
| | | Mean | Median | Mean | Median | Mean | Median | Mean | Median |
| n = 1000 | FRAGDIFF-T | **49.39** | **46.66** | **0.7928** | **0.7844** | **33.84** | **21.23** | **1.0455** | **0.9823** |
| | w/o CM | 47.68 | 45.45 | 0.8101 | 0.7861 | 33.39 | 20.93 | 1.0526 | 1.0056 |
| | TorDiff | 34.60 | 17.23 | 0.8933 | 0.8909 | 20.84 | 5.56 | 1.1897 | 1.1795 |
| | w/o CM | 45.39 | 39.44 | 0.8190 | 0.8040 | 28.74 | 15.00 | 1.1033 | 1.0667 |
| n = 10000 | FRAGDIFF-T | **62.82** | **69.70** | **0.6736** | **0.6505** | **43.10** | **37.72** | **0.9081** | **0.8840** |
| | w/o CM | 43.95 | 36.43 | 0.8353 | 0.8224 | 29.18 | 14.22 | 1.1038 | 1.0426 |
| | TorDiff | 52.76 | 54.10 | 0.7507 | 0.7379 | 33.88 | 20.64 | 1.0458 | 1.0371 |
| | w/o CM | 43.15 | 36.36 | 0.8478 | 0.8291 | 28.11 | 13.06 | 1.1051 | 1.0676 |
| Full | FRAGDIFF-T | **70.07** | **78.35** | **0.6092** | **0.5876** | **52.87** | **54.17** | **0.8003** | **0.7486** |
| | w/o CM | 37.52 | 25.00 | 0.8866 | 0.8863 | 23.73 | 8.22 | 1.1598 | 1.1307 |
| | TorDiff | 67.49 | 75.81 | 0.6339 | 0.6178 | 49.53 | 47.16 | 0.8269 | 0.7782 |
| | w/o CM | 34.99 | 20.88 | 0.9326 | 0.9174 | 23.08 | 8.13 | 1.1803 | 1.1340 |

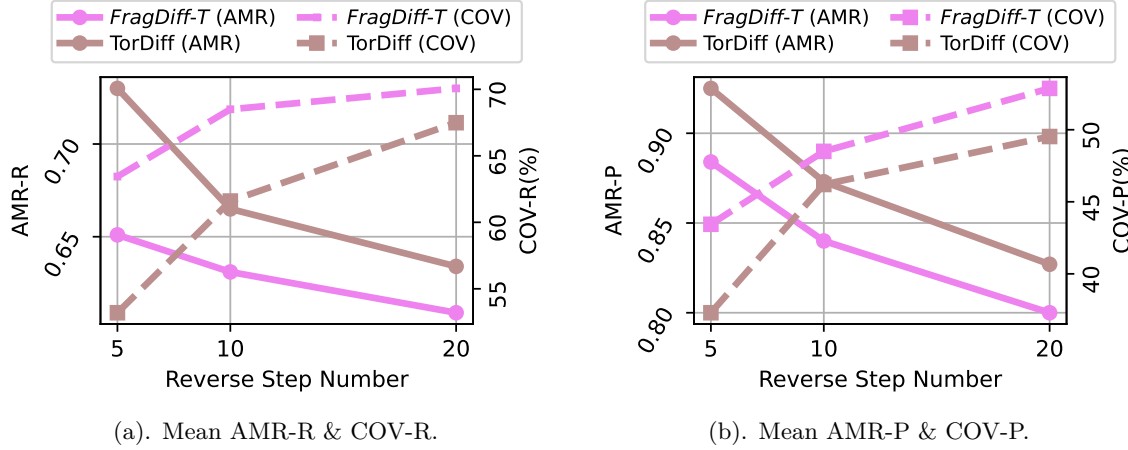

(a). Mean AMR-R & COV-R.  (b). Mean AMR-P & COV-P.

Figure 3: Reverse steps *v.s.* generation quality.

**Conformer Matching Ablation**   Table 7 presents an ablation study analyzing the impact of Conformer Matching during training on the GEOM-DRUGS test set. It reveals several key observations regarding the impact of Conformer Matching during training on the GEOM-DRUGS test set. Generally, models trained with CM outperform those without it, achieving higher Coverage percentages (COV-R and COV-P) and lower Average Minimum RMSD values (AMR-R and AMR-P).

However, an intriguing phenomenon occurs with TorDiff at $\mathbf{n} = 1000$: training without CM yields better performance than training with CM (mean COV-R of 45.39% vs. 34.60%, and mean AMR-R of 0.8190Å *v.s.* 0.8933Å). This suggests that, with limited data, directly using actual conformer structures for training may enhance TorDiff's generalization ability more than CM. As the dataset size increases, this advantage diminishes, and models trained without CM exhibit declining performance. This inverse relationship indicates that training on actual conformer data without CM may lead to overfitting to specific conformers, hampering generalization to unseen data as the model becomes more specialized on the training set. Conversely, CM helps prevent distributional shifts by aligning training and inference conformer distributions, which becomes increasingly beneficial with larger datasets. FRAGDIFF-T consistently outperforms TorDiff when CM is applied, suggesting it more effectively leverages CM for improved conformer generation. In Fig. 3, we further vary the number of steps used in the reverse diffusion process on GEOM-DRUGS for TorDiff and FRAGDIFF-T, showing that FRAGDIFF-T can generate better quality conformers under fewer steps. We provide additional experimental results, visualizations, and statistics in Appendix D.

## 5 Conclusion

In this work, we introduced Fragment-Augmented Diffusion (FRAGDIFF), a novel approach that incorporates traditional chemical fragmentation techniques into the pre-training phase of diffusion-based generative models. By decomposing molecules into smaller, meaningful fragments, FRAGDIFF enables the diffusion process to learn enhanced local geometry patterns while preserving global molecular topology. This data-centric strategy allows FRAGDIFF to bypass the need for complex diffusion model designs, offering a complementary pathway to improve conformer generation. Our comprehensive experiments across diverse datasets reveal that FRAGDIFF consistently surpasses state-of-the-art methods, especially in scenarios characterized by limited data availability and tasks involving the generation of large molecules. Overall, FRAGDIFF bridges traditional computational chemistry and modern diffusion models, enhancing conformational exploration and offering promising applications in drug discovery and materials science.

## Acknowledgments

This work is supported by a grant from ChemLex Technology Co., Ltd.. We would like to express our sincere gratitude to the editor and reviewers for their constructive comments and suggestions.

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

# A Supplementary Theroms and Proofs

## A.1 Diffusion Frameworks and Equivariance.

In the field of molecular conformer generation, diffusion models are employed to model the distribution of molecular structures, fully accounting for the inherent symmetry and degrees of freedom of molecules. Methods like GeoDiff Xu et al. (2022) operate directly on the Cartesian coordinates $\{\mathbf{x}_i\}_{i=1}^n$ of atoms, where each conformer $C$ is represented as a set of 3D points corresponding to the atomic positions $\mathbf{x}_i \in \mathbb{R}^3$. GeoDiff utilizes a diffusion process on these coordinates by adding noise to the atomic positions and then learning to denoise them using an $SE(3)$-equivariant graph neural network. Specifically, it updates the node features $\mathbf{h}_i$ and coordinates $\mathbf{x}_i$ through message passing over the molecular graph $\mathcal{G}$. The messages are computed based on the node features, edge features $e_{i,j}$, and pairwise distances $\|\mathbf{x}_i - \mathbf{x}_j\|$. The score function $\mathbf{s}_\theta(\{\mathbf{x}_i\}, t)$ predicts $SE(3)$-invariant scalar quantities for each chemical bond, which are used to update the positions:

$$\mathbf{x}_i^{l+1} = \mathbf{x}_i^l + \sum_{j \in \mathcal{N}(i)} \phi\left(\mathbf{h}_i^l, \mathbf{h}_j^l, \|\mathbf{x}_i^l - \mathbf{x}_j^l\|, e_{i,j}; \theta\right),$$

where $\phi$ is a function parameterized by $\theta$ that aggregates the information from neighboring atoms $\mathcal{N}(i)$. This approach ensures that the model's predictions are equivariant to rotations and translations, simplifying the learning process by making the predictions independent of the molecule's global position and orientation.

TorDiff Jing et al. (2022) focuses on the intrinsic torsional degrees of freedom of molecules. Rather than explicitly defining each torsion angle $\tau_i$, it leverages the fact that adjusting $\tau_i$ can be directly applied to the 3D atomic coordinates. Geometrically, changing a torsion angle corresponds to rotating a portion of the molecule around a chemical bond in three-dimensional space. For a rotatable chemical bond $(b_i, c_i)$, this rotation can be represented as:

$$\mathbf{x}'_{v(c_i)} = R\left(\theta\,\hat{\mathbf{b}}_{c_i}, \mathbf{x}_{c_i}\right)\mathbf{x}_{v(c_i)},$$

where $R\left(\theta\,\hat{\mathbf{b}}_{c_i}, \mathbf{x}_{c_i}\right) \in SE(3)$ is the rotation matrix corresponding to a rotation of angle $\theta$ around the axis $\hat{\mathbf{b}}_{c_i} = \dfrac{\mathbf{x}_{c_i} - \mathbf{x}_{b_i}}{\|\mathbf{x}_{c_i} - \mathbf{x}_{b_i}\|}$ passing through point $\mathbf{x}_{c_i}$, and $\mathbf{x}_{v(c_i)}$ are the positions of atoms bonded to atom $c_i$. Through this method, TorDiff directly manipulates the molecular geometry in three-dimensional space, simplifying the conformer generation process by focusing on internal rotational degrees of freedom.

Both methods consider an important symmetry: the invariance of physical energy under spatial inversion (mirror reflection), which is a fundamental property in physical systems Quack (2002). This implies that the learned probability density should satisfy $p(C) = p(-C)$, where $-C = \{-\mathbf{x} \mid \mathbf{x} \in C\}$. For the conditional distribution of conformers, this symmetry enforces consistency between original and inverted coordinates: $p(C \mid \mathcal{G}) = p(-C \mid \mathcal{G})$. Therefore, their score functions are designed to be invariant under $SE(3)$ transformations but change sign under spatial inversion, outputting pseudoscalars in $\mathbb{R}^m$. Mathematically, for all diffusion times $t$, this property can be expressed as:

$$\nabla_C \log p_t(C \mid \mathcal{G}) = -\nabla_C \log p_t(-C \mid \mathcal{G}),$$

where $\mathcal{G}$ denotes the molecular graph structure. By incorporating these symmetries into the model architecture, the models enhance their generalization ability, ensuring that predictions are physically meaningful and adhere to the fundamental symmetries of molecular systems Xu et al. (2022).

## B  Fragmentation-Based Decomposition and Error Analysis

### B.1  Fragmentation Methods

To validate the effectiveness and differences of various decomposition methods in learning conformer structures in the fragmentation torsion space, we analyzed several fragmentation rules in FRAGDIFF-T implementations. These methods provide domain knowledge and deeper insights into the task. The analyzed rules used include the BRICS method and RECAP method, and a graph-based fragmentation method can be employed by analyzing the connectivity of molecular graphs, which is also the method that Torsional Diffusion used for selecting the rotatable bond (Gordon et al., 2012; Stern et al., 2020; Jing et al., 2022), this method identifies cut edges by examining whether the removal of an edge disconnects the graph into separate components.

**RECAP (Retrosynthetic Combinatorial Analysis Procedure)**  (Lewell et al., 1998) RECAP is a classical technique aimed at decomposing complex molecules into smaller, manageable fragments through retrosynthetic analysis. The core principle involves identifying and cleaving chemical bonds that are common in organic synthesis, such as ester, amide, and ether bonds. These bonds are selected based on their prevalence and the ease of cleavage, prioritizing those that connect functional groups to generate fragments with clear chemical functionalities.

**BRICS (Breaking of Retrosynthetically Interesting Chemical Substructures)**  (Degen et al., 2008) BRICS applies a comprehensive set of rules to identify and cleave key substructures in chemical compounds, considering not just bond types but also the surrounding chemical environment, such as aromaticity and heterocycles. This allows BRICS to generate complex and diverse fragments, supporting multi-functional group cleavage to produce synthetically feasible and biologically relevant fragments.

---

**Algorithm 1** Graph-based Molecular Fragmentation.

---

**Require:** Molecular structure $M$
**Ensure:** Set of rotatable bonds $R$, Set of fragments $F$

1:   $\mathcal{C} \leftarrow \text{ConvertToGraph}(M)$          ▷ Convert molecular structure to undirected graph
2:   $R \leftarrow \emptyset$          ▷ Initialize set of rotatable bonds
3:   $F \leftarrow \emptyset$          ▷ Initialize set of fragments
4:   **for** each edge $e \in \mathcal{C}.edges$ **do**
5:      $\mathcal{C}_{temp} \leftarrow \mathcal{C} \setminus \{e\}$          ▷ Temporarily remove edge
6:      $C \leftarrow \text{GetConnectedComponents}(\mathcal{C}_{temp})$
7:      **if** $|C| > 1$ **and** $\forall c \in C : |c| \geq 2$ **then**          ▷ Check disconnection and size
8:         $R \leftarrow R \cup \{e\}$          ▷ Add to rotatable bonds
9:         $F \leftarrow F \cup C$          ▷ Add components as fragments
10:      **end if**
11:   **end for**
12:   **return** $R, F$

---

**Graph-based Fragmentation**   (Gordon et al., 2012; Stern et al., 2020; Jing et al., 2022) We consider a bond freely rotatable if severing the bond creates two connected components of the total graph, each of which has at least two atoms. It guarantees that torsion angles in cycles (or rings), which cannot be rotated independently, are considered part of the local structure. It can be described as in Algorithm. 1 Thus, it allows for the identification of edges that, when removed, split the molecular graph into meaningful substructures. The algorithm ensures that fragments retain their connectivity, making it particularly useful for identifying torsion-related substructures.

## B.2   Fragmentation Augmentation Error Analysis

In fragment-based augmentation, breaking down molecules into smaller components introduces an approximation error, $\epsilon$, between the true molecular coordinates, $C$, and the reconstructed coordinates, $\hat{C}$. This error results from the loss of structural and conformational information, influenced by factors like the selection of cutting edges and disruption of chemical interactions. To assess this error's impact on retaining information about the original molecule, we analyze the mutual information, $I(\hat{C}; C)$, between $\hat{C}$ and $C$. This is expressed as $I(\hat{C}; C) = H(C) - H(C \mid \hat{C})$, where $H(C)$ is the entropy of the molecular coordinates, and $H(C \mid \hat{C})$ is the uncertainty in $C$ given $\hat{C}$. Since $\hat{C}$ is a perturbed version of $C$, this conditional entropy equals the entropy of the error $\epsilon$. Assuming $\epsilon$ follows a zero-mean multivariate Gaussian distribution with covariance matrix $\Sigma$, its differential entropy is $H(\epsilon) = \frac{1}{2} \ln \left( (2\pi e)^k \det \Sigma \right)$, where $k$ is the dimensionality of the molecular coordinate space. The error $\epsilon$ reduces the mutual information between $\hat{C}$ and $C$. As $\epsilon$ increases, mutual information decreases, meaning less information about $C$ can be inferred from $\hat{C}$. Therefore, minimizing fragmentation errors is crucial for improving molecular modeling and conformation prediction. By quantifying $\epsilon$'s impact on mutual information, we can compare different fragmentation strategies. Strategies that minimize $\epsilon$ preserve more mutual information, leading to more accurate predictions. This highlights the importance of choosing fragmentation methods that retain as much structural and conformational information as possible.

**Energy Perspective**   Building upon the previous analysis of fragmentation errors and their impact on mutual information, we now examine these effects from an energy perspective. In molecular simulations, the force field (FF) describes the potential conformational energy $E_{\text{conf}}$ of a molecular system, using mathematical expressions and associated parameters to model both bonded and non-bonded interactions. The accuracy of the estimated conformation $\hat{C}$ is intrinsically linked to how well these energy terms are captured during the fragmentation process. Bonded interactions account for atoms connected by chemical bonds and include bond stretching, angle bending, and dihedral (torsional) rotations:

$$E_{\text{bonded}} = \sum_{\text{bonds}} k_b (b - b_0)^2 + \sum_{\text{angles}} k_\theta (\theta - \theta_0)^2 + \sum_{\text{dihedrals}} \sum_n \frac{V_n}{2} [1 + \cos(n\phi - \gamma_n)], \tag{1}$$

where the bond stretching term is modeled using a harmonic potential with $k_b$ representing the bond force constant, $b$ the bond length, and $b_0$ the equilibrium bond length; angle bending is similarly described using $k_\theta$ as the angle force constant, $\theta$ as the bond angle, and $\theta_0$ as the equilibrium bond angle; and dihedral rotation is expressed as a Fourier series expansion with $V_n$ denoting the torsional barrier amplitude, $n$ the periodicity, $\phi$ the dihedral angle, and $\gamma_n$ the phase offset. Non-bonded interactions consider pairs of atoms not directly bonded and include Van der Waals forces and electrostatic interactions:

Figure 4: Visualization of energy-based analysis.

$$E_{\text{nonbonded}} = \sum_{i<j} \left[ 4\varepsilon_{ij} \left( \left( \frac{\sigma_{ij}}{r_{ij}} \right)^{12} - \left( \frac{\sigma_{ij}}{r_{ij}} \right)^6 \right) + \frac{q_i q_j}{4\pi\varepsilon_0 \varepsilon_r r_{ij}} \right], \quad (2)$$

where the Van der Waals interactions are described by the Lennard-Jones potential—dependent on $\varepsilon_{ij}$ (the depth of the potential well), $\sigma_{ij}$ (the finite distance at which the interparticle potential is zero), and $r_{ij}$ (the distance between atoms $i$ and $j$)—and the electrostatic interactions are modeled using the Coulombic potential with $q_i$ and $q_j$ as the partial charges, $\varepsilon_0$ the vacuum permittivity, and $\varepsilon_r$ the relative permittivity (dielectric constant). To minimize the error variance $\sigma^2$ and maximize the mutual information $I(\hat{C}; C)$ between the fragment conformation $C$ and the full molecular conformation $\hat{C}$, fragmentation methods must carefully consider several key factors. Each factor affects the molecular conformational properties by influencing the potential energy surface and conformational distributions (Stern et al., 2020; Horton et al., 2022; Stern et al., 2022).

**Discussions on the Effectivenesses of Fragmentation Methods** Fragmentation plays a crucial role in molecular simulations by altering both bonded and non-bonded interactions, which in turn affects the conformational energy landscape. When molecular bonds are severed to create fragments, the local chemical environment is modified, impacting energy parameters and the balance of forces that dictate molecular conformations. For bonded interactions, fragmentation influences torsional energy terms, such as barrier heights ($V_n$) and phase offsets ($\gamma_n$) associated with dihedral angles ($\phi$). This is particularly significant when fragmentation disrupts conjugation or resonance effects (Horton et al., 2022; Stern et al., 2022). For instance, cutting bonds in conjugated systems or aromatic rings interrupts electron delocalization, altering the torsional potential energy surface. These changes can be quantified by comparing the electron density distributions of the fragment ($\rho_{\text{frag}}(\mathbf{r})$) and the full molecule ($\rho_{\text{full}}(\mathbf{r})$), where $\mathbf{r}$ represents the spatial positions of the electrons. Additionally, removing bulky substituents near torsional bonds reduces steric hindrance, lowering torsional barriers and shifting equilibrium angles. This results in deviations between the conformations of the fragment ($C$) and the full molecule ($\hat{C}$), increasing error variance and reducing the mutual information $I(\hat{C}; C)$ between the fragment conformation ($C$) and the full molecular conformation ($\hat{C}$). For non-bonded interactions, fragmentation affects the balance of Van der Waals and electrostatic forces, which are essential for conformational preferences (Stern et al., 2022). The removal of atoms and groups decreases the number of atom pairs contributing to Van der Waals interactions, especially those spanning the fragmentation site, altering the balance of attractive and repulsive forces. Similarly, the deletion or modification of charged or polar groups impacts electrostatic interactions by changing the distribution of partial charges ($q_i$ and $q_j$). Disruption of hydrogen bonds and other electrostatic interactions can significantly reshape the energy landscape, particularly in systems dominated by Coulombic potentials, which are highly sensitive to the spatial arrangement of charged species.

**Bridging the Analysis with Experimental Results** The experimental data presented in Table 6 corroborate our theoretical analysis regarding the impact of fragmentation methods on approximation errors and mutual information $I(\hat{C}_b; C)$. The comprehensive FRAGDIFF-T model, which integrates graph-based fragmentation using both BRICS and RECAP edges, demonstrates superior performance across all metrics. It achieves the highest mean COV-R (51.17%) and COV-P (50.10%), alongside the lowest AMR-R and AMR-P values. Excluding BRICS edges results in a decline in precision metrics, with COV-P decreasing

to 34.51% and AMR-P rising to 1.0461 Å, underscoring the critical role of BRICS fragmentation in enhancing precision during conformer generation. Similarly, the omission of RECAP edges negatively impacts recall metrics, as evidenced by a reduction in COV-R to 49.38% and an increase in AMR-R to 0.7609 Å, highlighting the importance of RECAP fragmentation for achieving comprehensive conformational coverage. The most pronounced performance degradation occurs when both BRICS and RECAP edges are removed, resulting in the lowest COV-R (48.60%) and COV-P (46.89%), as well as the highest AMR-R and AMR-P values. This emphasizes the complementary nature of BRICS and RECAP in maintaining essential molecular structural features for accurate conformer generation. These results validate our theoretical framework, suggesting that **optimal fragmentation strategies** that maximize mutual information $I(\hat{C}_b; C)$ and minimize approximation errors are pivotal for enhancing model performance. Therefore, selecting fragmentation strategies that align with theoretical insights—particularly those preserving chemical properties as defined in Section B.1—is vital for optimizing model efficacy in practical applications.

## C  Reproducbility

**Experimental Details**   For the FRAGDIFF-T pretraining and finetuning phases, we adopted the setup used in Torsional Diffusion (Jing et al., 2022). Specifically, for conformer generation on GEOM-DRUGS, we mainly followed the setup used in Jing et al. (2022). We trained the Torsional Diffusion models on NVIDIA RTX A100 GPUs for 250 epochs using the Adam optimizer for GEOM-DRUGS and GEOM-QM9. The primary hyperparameters were optimized using the validation set, resulting in the following configurations: an initial learning rate of 0.001, a learning rate scheduler with a patience of 20 epochs, 4 network layers, a second-order maximum representation, a cutoff radius $r_{\max}$ of 10 Å, and the inclusion of batch normalization. Specifically, following the setup used in (Jing et al., 2022), we used the model trained from GEOM-DRUGS for GEOM-XL evaluation. The results reported for FRAGDIFF-T utilize 20 reverse diffusion steps, consistent with the approach in Jing et al. (2022). The minimum fragment size $z$ was set to 10 for both GEOM-DRUGS and GEOM-XL, while no such limit was applied in the GEOM-QM9 experiments. The maximum fragmentation edge number $\kappa$ is set to 5 for all datasets. For FRAGDIFF-G pretraining and finetuning, we employed the same setup as GeoDiff (Xu et al., 2022). This ensures that our approach is consistent with existing methodologies while allowing for a fair comparison of results across different models and datasets. Consistent with the approach in Jing et al. (2022), for each molecule that has $K$ ground truth conformers, we generate 2000 conformers. The datasets were randomly divided into training, validation, and test sets with sizes as follows: for GEOM-DRUGS, there are 243,473 training samples, 30,433 validation samples, and 1,000 test samples; for GEOM-QM9, there are 106,586 training samples, 13,323 validation samples, and 1,000 test samples. Since GEOM-XL is used solely for testing, its test set includes all 102 molecules from the MoleculeNet dataset that contain at least 100 atoms. In Torsional Diffusion (Jing et al., 2022), an effective design is introduced for training generative models in torsional space, which we adopt in our work. Recognizing that the set of possible stable local structures $L$ for a given molecule is highly constrained and can be accurately predicted using fast cheminformatics methods like RDKit ETKDG (Riniker & Landrum, 2015), we use RDKit to provide approximate samples from $p_\theta(L)$. This allows us to focus on developing a diffusion-based generative model to learn the distribution $p_\theta(\tau \mid L)$ over torsion angles $\tau$, conditioned on the given graph and local structure. To enhance the performance of our model, we employ a conformer matching procedure as described in Jing et al. (2022), where training on synthetic conformers produced by conformer matching has shown significantly better results than using ground truth conformers alone. Specifically, for a molecule with $K$ conformers, we generate $K$ random local structure estimates $\hat{L}$ using RDKit. To align these estimates with the ground truth conformers $C$, we compute a $K \times K$ cost matrix, where each entry represents the lowest Root Mean Square Deviation (RMSD) achievable by adjusting the torsion angles of $\hat{L}$ to match $C$. Solving the linear sum assignment problem on this cost matrix (Crouse, 2016; Stärk et al., 2022), we find the optimal matching between the true conformers $C$ and the estimates $\hat{C}$. For each matched pair, we refine the alignment by performing a differential evolution optimization over the torsion angles to obtain the optimal conformer $\hat{C}$ (Méndez-Lucio et al., 2021). This comprehensive matching and refinement process ensures consistency between the local structures seen during training and inference, effectively preventing any distributional shift.

---

**Algorithm 2** FRAGDIFF Training and Inference

---

**Input:** Molecular graphs $\{\mathcal{G}_0, \ldots, \mathcal{G}_N\}$ with ground truth conformers $\{C_{\mathcal{G}_0,1}, \ldots, C_{\mathcal{G}_N,K}\}$;
    learning rate $\alpha$; number of conformers $K$; diffusion steps $T$;
    maximum selected edges $\kappa$; minimum fragment size $z$.

**Output:** Trained score model $\mathbf{s}_\theta$; generated conformers $\{C_1, \ldots, C_K\}$.

    **Pretraining Phase:**
1: Initialize augmented training set $\mathcal{F} \leftarrow \emptyset$
2: **for all** molecular graph $\mathcal{G} \in \{\mathcal{G}_0, \ldots, \mathcal{G}_N\}$ **do**
3:     Identify cut-edges $\mathcal{E}_{\text{cut}} \subseteq \mathcal{E}$ in $\mathcal{G}$
4:     Let $B = |\mathcal{E}_{\text{cut}}|$ be the total number of cut-edges
5:     Select $\tilde{K} = \min(B, \kappa)$ edges from $\mathcal{E}_{\text{cut}}$
6:     Decompose $\mathcal{G}$ into $B + 1$ fragments $\{\hat{\mathcal{G}}_b\}_{b=1}^{B+1}$
7:     Remove fragments with $|\mathcal{V}(\hat{\mathcal{G}}_b)| < z$
8:     Add remaining fragments to $\mathcal{F}$ with $\hat{C}_b \leftarrow \text{ExtractSubspace}(C_{\mathcal{G},k})$
9: **end for**
10: $\text{TRAIN}(\mathcal{F})$
    **Finetuning Phase:**
11: $\text{TRAIN}(\{\mathcal{G}_0, \ldots, \mathcal{G}_N\})$
    **Training Procedure:**
12: **function** $\text{TRAIN}(\mathcal{D})$
13:     **for** epoch $= 1$ **to** epoch$_{\max}$ **do**
14:         **for all** $\mathcal{G} \in \mathcal{D}$ **or** $\hat{\mathcal{G}}_b \in \mathcal{D}$ **do**
15:             Sample $t \sim \text{Uniform}\{1, \ldots, T\}$
16:             Sample $C^0 \sim \{C_{\mathcal{G},1}, \ldots, C_{\mathcal{G},K}\}$
17:             Compute $\alpha_t = \prod_{s=1}^{t}(1 - \beta_s)$
18:             Sample $\epsilon \sim \mathcal{N}(0, \mathbf{I})$
19:             Perturb $C^t = \sqrt{\alpha_t}C^0 + \sqrt{1 - \alpha_t}\epsilon$
20:             Compute $\mathbf{s}_\theta = \mathbf{s}_\theta(\mathcal{G}, C^t, t)$
21:             Calculate $\mathcal{L} = \mathbb{E}\left[\|\mathbf{s}_\theta - \nabla_{C^t} \log p_{t|0}(C^t|C^0)\|_2^2\right]$
22:             Update $\theta \leftarrow \theta - \alpha\nabla_\theta\mathcal{L}$
23:         **end for**
24:     **end for**
25: **end function**
    **Inference Procedure:**
26: **for all** molecular graph $\mathcal{G}$ **do**
27:     Initialize $C^T \sim \mathcal{N}(0, \mathbf{I})$
28:     **for** $t = T$ **downto** 1 **do**
29:         Compute $\mu_\theta = \frac{1}{\sqrt{\alpha_t}}\left(C^t - \frac{\beta_t}{\sqrt{1-\alpha_t}}\mathbf{s}_\theta(\mathcal{G}, C^t, t)\right)$
30:         Sample $C^{t-1} \sim \mathcal{N}(\mu_\theta, \sigma_t^2\mathbf{I})$
31:     **end for**
32:     Output $C^0$ as generated conformer
33: **end for**

---

# D   Further Results and Statistics

## D.1   Further Visualizations on Conformation Generation for Large Molecules (GEOM-XL)

Table 8 provides additional visualizations of conformer generation results on the GEOM-XL dataset, focusing on large molecules. These examples complement our earlier discussion on the superior generalization performance of FRAGDIFF-T on large molecules. The table compares the generated conformers from FRAGDIFF-T and TorDiff with the reference structures. It shows that FRAGDIFF-T produces conformers that closely resemble the reference structures on given examples, further elucidates the performance improvements pre-

Table 8: Visualizations on conformer generation examples on GEOM-XL.

| Graph | Reference | FRAGDIFF-T | TorDiff |
|---|---|---|---|

Figure 5: Average Fragment Number *v.s.* Minimum Fragment Size $z$ on different fragmentation methods.

sented in Table 3, highlighting FRAGDIFF-T's exceptional ability to handle large and complex molecules by generating conformers that closely match the reference structures.

## D.2 Further Fragmentation Statistics

Figure 5 illustrates how the minimum fragment size $z$ affects the average number of fragments per molecule for the Graph-based, BRICS, and RECAP fragmentation methods across the GEOM-QM9, GEOM-DRUGS, and GEOM-XL datasets. GEOM-QM9, comprising small molecules averaging 11 atoms, shows that the Graph-based method generates significantly more fragments when $z$ is small, but the fragment count drops rapidly as $z$ increases due to the limited molecular size. BRICS and RECAP produce fewer fragments with less sensitivity to $z$ changes. In the GEOM-DRUGS dataset, with molecules averaging 44 atoms,

Table 9: RMSD comparison between extracted and energy-minimized fragment conformations.

| Method | Mean RMSD (Å) | Median RMSD (Å) |
|--------|---------------|-----------------|
| BRICS | 0.4935 | 0.3962 |
| RECAP | 0.5380 | 0.4371 |
| Graph | 0.4673 | 0.3753 |

all methods produce more fragments, but the Graph-based method still leads, and the decline in fragment numbers with increasing $z$ is more gradual. For the GEOM-XL dataset, containing large molecules averaging 132 atoms, all methods generate a higher number of fragments, and the differences between methods become less pronounced as $z$ increases. The Graph-based method remains the most sensitive to changes in $z$, while BRICS and RECAP display steady decreases. Overall, these trends highlight that larger molecules permit more fragmentation, and the Graph-based method consistently yields more fragments, especially at smaller $z$ values, whereas BRICS and RECAP are less influenced by the minimum fragment size due to their inherent fragmentation rules.

### D.3 Further Experiments on Fragment Conformational Validity

To validate our assumption that fragment conformations extracted directly from larger molecules ($\hat{C}_b$) provide reasonable approximations of their relaxed, isolated conformations ($C_b$), we conducted a empirical analysis comparing extracted fragments with their energy-minimized structures. This analysis helps quantify the approximation error inherent in our fragment-based augmentation approach. We randomly sampled 1000 molecules from the GEOM-DRUGS dataset, selecting one conformer from each molecule for testing and analysis. For each fragment generated through our different fragmentation methods (BRICS, RECAP, and graph-based). we performed energy minimization using the MMFF94s force field implemented in RDKit to obtain relaxed conformations. We then calculated the Root Mean Square Deviation (RMSD) between the original extracted conformations and their corresponding minimized structures. The results are summarized in Table 9.

The results reveal that the geometric deviation between extracted and relaxed fragment conformations is relatively small, with median RMSD values consistently below 0.5 Å. This is particularly noteworthy as these deviations are well below the threshold of 0.75 Å used in our GEOM-DRUGS evaluations for considering conformers similar. The graph-based method shows the smallest deviation (median RMSD = 0.3753 Å), followed by BRICS (0.3962 Å) and RECAP (0.4371 Å). These findings support our theoretical analysis in Section B, suggesting that our fragmentation methods effectively preserve local chemical environments and structural features. The small RMSD values indicate that fragments extracted from larger molecules maintain reasonable geometric validity, justifying their use as training data without the need for additional relaxation steps. This conformational stability likely contributes to the effectiveness of our fragment-based augmentation strategy in improving conformer generation performance.

We further provide visual comparisons between the original and energy-minimized conformations for several fragments obtained specifically using the graph-based fragmentation method, as illustrated in Figure 6, demonstrating the minimal structural changes that occur during energy minimization. As shown in the examples, even for diverse molecular structures ranging from simple ring systems (Example 1) to more complex heterocyclic compounds (Examples 7-10), the extracted fragments maintain their structural integrity well after energy minimization, further validating the reliability of fragment-augmented pretraining.

### D.4 Further Results on Boltzmann generation

For Boltzmann generation experiments, we follow the experimental setup from Jing et al. (2022) to evaluate FRAGDIFF-T. Compared to TorDiff and annealed importance sampling (AIS), FRAGDIFF-T achieves superior effective sample size (ESS) across temperatures[1] (Table 10), demonstrating enhanced sampling fidelity.

---

[1]The comparison focuses on the effective sample size of 32 samples per molecule, which quantifies how closely the generated samples match the true Boltzmann distribution.

| Example 1 | | | | Example 2 | | | |
|---|---|---|---|---|---|---|---|
| Complete | Fragment | Minimized | RMSD | Complete | Fragment | Minimized | RMSD |
| | | | 0.2568 | | | | 0.3652 |

| Example 3 | | | | Example 4 | | | |
|---|---|---|---|---|---|---|---|
| Complete | Fragment | Minimized | RMSD | Complete | Fragment | Minimized | RMSD |
| | | | 0.3384 | | | | 0.4554 |

| Example 5 | | | | Example 6 | | | |
|---|---|---|---|---|---|---|---|
| Complete | Fragment | Minimized | RMSD | Complete | Fragment | Minimized | RMSD |
| | | | 0.4662 | | | | 0.5778 |

| Example 7 | | | | Example 8 | | | |
|---|---|---|---|---|---|---|---|
| Complete | Fragment | Minimized | RMSD | Complete | Fragment | Minimized | RMSD |
| | | | 0.3365 | | | | 0.4033 |

| Example 9 | | | | Example 10 | | | |
|---|---|---|---|---|---|---|---|
| Complete | Fragment | Minimized | RMSD | Complete | Fragment | Minimized | RMSD |
| | | | 0.1412 | | | | 0.3348 |

Figure 6: Visualization of fragment conformational stability. For each example, we show the complete molecule (left), the fragment extracted using the graph-based method (middle), and its energy-minimized version (right). RMSD values quantify the structural differences between extracted and energy-minimized fragments. These visualizations demonstrate that fragments maintain their structural integrity with minimal geometric changes after energy minimization.

Table 10: Comparison of different methods at various temperatures.

| Method | Temp. (K) | | |
|---|---|---|---|
| | 1000 | 500 | 300 |
| Uniform | 1.71 | 1.21 | 1.02 |
| AIS | 3.12 | 1.76 | 1.30 |
| TorDiff | 11.42 | 6.42 | 4.68 |
| FRAGDIFF-T | **11.65** | **6.78** | **5.02** |

At 300K, it attains an ESS of 5.02 (*v.s.* TorDiff's 4.68), with consistent improvements at higher temperatures. The results validate that fragment-based diffusion better captures the true Boltzmann distribution than global sampling approaches.

## D.5 Pretraining-Only Model Performance

To validate the effectiveness of our training approach, we conducted an ablation study comparing our full FRAGDIFF-T model with a variant that uses only fragment pretraining without the subsequent finetuning stage. This experiment helps isolate the specific contribution of each stage in our training pipeline.

Table 11: Comparison of pretraining-only versus full FRAGDIFF-T model on the GEOM-DRUGS test set with threshold $\delta = 0.75$ Å(Mean).

| Models | COV-R (%) ↑ | | AMR-R (Å) ↓ | | COV-P (%) ↑ | | AMR-P (Å) ↓ | |
|---|---|---|---|---|---|---|---|---|
| | Mean | Median | Mean | Median | Mean | Median | Mean | Median |
| Pretraining-only | 44.12 | 37.12 | 0.849 | 0.816 | 33.09 | 20.67 | 1.047 | 1.011 |
| FRAGDIFF-T | **70.07** | **78.35** | **0.609** | **0.588** | **52.87** | **54.17** | **0.800** | **0.749** |

Table 12: Comparison of torsional angle errors between TorDiff and FRAGDIFF-T on the GEOM-DRUGS dataset.

| Method | Mean Torsional Error | Median Torsional Error |
|---|---|---|
| TorDiff | 0.2645 | 0.2709 |
| FRAGDIFF-T | **0.2562** | **0.2618** |

We trained a model using only the fragment pretraining stage on the GEOM-DRUGS dataset, maintaining identical hyperparameters, network architecture, and training iterations as used in our full FRAGDIFF-T model. During inference, this pretraining-only model was directly applied to generate conformers for the test set molecules without any molecule-level finetuning. Table 11 presents the performance comparison between the pretraining-only model and our full FRAGDIFF-T model on the GEOM-DRUGS test set with threshold $\delta = 0.75$ Å.

The results demonstrate that while fragment pretraining alone provides reasonable performance, the full approach with subsequent finetuning substantially outperforms the pretraining-only variant across all metrics. Specifically, the full FRAGDIFF-T model achieves approximately 59% higher COV-R (70.07% vs. 44.12%) and 60% higher COV-P (52.87% vs. 33.09%) compared to the pretraining-only model. Similarly, the full model reduces AMR-R by 28% (0.609 Å vs. 0.849 Å) and AMR-P by 24% (0.800 Å vs. 1.047 Å).

These findings confirm that while fragment pretraining contributes significantly to model performance by helping the model learn generalizable patterns from fragment substructures, the molecule-level finetuning stage is essential for optimal conformer generation. The finetuning phase allows the model to effectively integrate the knowledge learned from fragments into whole-molecule contexts, accounting for long-range interactions and global molecular properties that cannot be fully captured through fragments alone.

### D.6 Further Results on Torsional Angle Error Measurements

In addition to the RMSD-based metrics (Coverage and AMR) commonly used in conformer generation evaluation, we also assessed the direct torsional angle accuracy of our models. Unlike global RMSD measurements that capture overall structural similarity, torsional angle errors provide insight into the model's ability to accurately predict local geometric features that determine molecular conformation.

For this analysis, we compared the torsional angles of generated conformers with the reference conformers from the ground truth conformers. The torsional error is calculated as the absolute difference between the generated and reference angles, accounting for the circular nature of angular measurements. Specifically, we identified all rotatable bonds (non-ring single bonds) in the molecules, calculated the dihedral angles formed by the four atoms across each rotatable bond, and measured the minimum angular difference between corresponding angles in the reference and predicted conformers.

Table 12 presents the mean and median torsional errors for both TorDiff and our FRAGDIFF-T approach on the GEOM-DRUGS dataset. The results show that FRAGDIFF-T achieves lower torsional angle errors compared to TorDiff, with improvements in both mean and median measurements. This indicates that our fragment-based pretraining approach enhances the model's understanding of local torsional preferences, complementing the improvements observed in the global RMSD-based metrics. The consistent enhancement across both local and global metrics further validates the effectiveness of our fragment augmentation strategy in improving conformer generation quality.

Table 13: Comparison between pretrain-finetune and joint loss approaches on the GEOM-DRUGS test set with $\delta = 0.75$ Å.

| Models | COV-R (%) ↑ | AMR-R (Å) ↓ | COV-P (%) ↑ | AMR-P (Å) ↓ |
|---|---|---|---|---|
| FRAGDIFF-T (Joint Loss) | 68.84 | 0.622 | 51.02 | 0.813 |
| FRAGDIFF-T (Pretrain-Finetune) | **70.07** | **0.609** | **52.87** | **0.800** |

Table 14: Quality of generated conformers for the GEOM-DRUGS test set in terms of Coverage (%) and Average Minimum RMSD (Å) with $\delta = 0.75$ Å.

| Models | COV-R (%) ↑ | | AMR-R (Å ↓) | | COV-P (%) ↑ | | AMR-P (Å ↓) | |
|---|---|---|---|---|---|---|---|---|
| | Mean | Median | Mean | Median | Mean | Median | Mean | Median |
| DMCG | 57.55 | 59.00 | 0.722 | 0.723 | 37.57 | 35.33 | 0.944 | 0.909 |
| DMCG-Frag | 56.50 | 57.76 | 0.726 | 0.726 | 37.11 | 34.07 | 0.944 | 0.910 |
| GeoDiff | 45.61 | 49.32 | 0.862 | 0.852 | 21.47 | 14.55 | 1.171 | 1.123 |
| FRAGDIFF-G | 51.56 | 52.34 | 0.847 | 0.838 | 26.13 | 20.63 | 1.095 | 1.084 |
| TorDiff | 67.49 | 75.81 | 0.634 | 0.618 | 49.53 | 47.16 | 0.827 | 0.778 |
| FRAGDIFF-T | **70.07** | **78.35** | **0.609** | **0.588** | **52.87** | **54.17** | **0.800** | **0.749** |

## D.7 Comparison with Joint Loss Training

Intuitively, our sequential approach offers several advantages: (1) it establishes robust representations of local chemical patterns before integrating them into global molecular contexts; (2) it prevents boundary artifacts at fragmentation sites from immediately affecting whole-molecule predictions; and (3) it creates a beneficial curriculum effect where the model first masters simpler fragment-level patterns before tackling more complex molecular relationships.

To empirically validate our design choice of using a sequential pretrain-finetune approach rather than a single-stage joint training method, we conducted additional experiments comparing these two methodologies. The joint loss approach simultaneously trains on both fragment data and complete molecules by averaging losses across both data types within each training batch, rather than using separate training phases.

Table 13 presents the performance comparison between our pretrain-finetune strategy and the joint loss approach. To ensure fair comparison, we allocated equivalent computational resources to both methods, with the joint loss model receiving twice the standard training resources to match the total computation of our two-phase approach. The results demonstrate that our sequential pretrain-finetune strategy consistently delivers better performance across all metrics compared to the joint loss approach. These empirical findings align with our theoretical understanding and provide further justification for our chosen methodology, particularly considering its enhanced generalization capabilities for larger molecules and in data-scarce scenarios as shown in Section 5 and Table 3.

## D.8 Comparison with Direct Molecular Conformation Generation

The main scope of this study discusses the fragmentation pretraining strategy as an enhancement for diffusion-based molecular conformation generation models. To further provide insights on whether similar benefits extend to non-diffusion approaches, we conducted additional experiments with DCMG (Zhu et al., 2022), a representative coordinate regression method that directly predicts 3D coordinates through iterative GNN blocks with specialized loss functions ensuring roto-translation and permutation invariance.

We implemented the fragment pretraining strategy for DMCG (denoted as DMCG-Frag) following the same procedure described in Section 3. Table 14 presents a comprehensive comparison of all methods on the GEOM-DRUGS dataset with $\delta = 0.75$ Å. Notably, our FRAGDIFF-T substantially outperforms DMCG across all metrics. While GeoDiff underperforms compared to DMCG, our FRAGDIFF-G enhancement significantly narrows this gap. Interestingly, when applying the fragmentation pretraining strategy to DMCG (DMCG-Frag), performance slightly decreases rather than improves, suggesting that the fragmentation approach particularly benefits diffusion-based models. Table 15 shows results on the GEOM-QM9 dataset with

Table 15: Quality of generated conformers for the GEOM-QM9 test set in terms of Coverage (%) and Average Minimum RMSD (Å) with $\delta = 0.5$ Å.

| Models | COV-R (%) ↑ | | AMR-R (Å ↓) | | COV-P (%) ↑ | | AMR-P (Å ↓) | |
|---|---|---|---|---|---|---|---|---|
| | Mean | Median | Mean | Median | Mean | Median | Mean | Median |
| DMCG | **96.3** | **100.0** | 0.206 | 0.200 | 87.2 | 91.0 | 0.287 | 0.292 |
| DMCG-Frag | **96.3** | **100.0** | 0.214 | 0.215 | 88.7 | 92.6 | 0.284 | 0.288 |
| GeoDiff | 90.1 | 93.4 | 0.209 | 0.198 | 52.8 | 50.3 | 0.445 | 0.427 |
| FRAGDIFF-G | 91.1 | 95.1 | 0.199 | 0.194 | 56.0 | 55.0 | 0.431 | 0.421 |
| TorDiff | 92.8 | **100.0** | 0.178 | 0.147 | **92.7** | **100.0** | **0.221** | 0.195 |
| FRAGDIFF-T | 93.2 | **100.0** | **0.175** | **0.139** | **93.1** | **100.0** | **0.218** | **0.189** |

Table 16: Quality of generated conformers for the GEOM-DRUGS test set with $\delta = 0.75$ Å on varying available training samples **n**. (Mean)

| Models | FRAGDIFF-G | | | | GeoDiff | | | |
|---|---|---|---|---|---|---|---|---|
| Metric | **COV-R** | **AMR-R** | **COV-P** | **AMR-P** | **COV-R** | **AMR-R** | **COV-P** | **AMR-P** |
| 1000 | 13.07 | 1.250 | 7.658 | 1.346 | 8.780 | 1.347 | 2.191 | 1.413 |
| 5000 | 15.69 | 1.159 | 10.03 | 1.251 | 13.83 | 1.242 | 6.327 | 1.383 |
| 10000 | 30.30 | 1.051 | 21.02 | 1.166 | 16.78 | 1.175 | 9.196 | 1.261 |
| Models | FRAGDIFF-T | | | | TorDiff | | | |
| Metric | **COV-R** | **AMR-R** | **COV-P** | **AMR-P** | **COV-R** | **AMR-R** | **COV-P** | **AMR-P** |
| 1000 | **49.39** | **0.793** | **33.84** | **1.046** | 34.60 | 0.983 | 20.84 | 1.190 |
| 5000 | **51.17** | **0.752** | **34.51** | **1.039** | 44.61 | 0.821 | 25.77 | 1.110 |
| 10000 | **62.82** | **0.674** | **43.10** | **0.908** | 52.76 | 0.751 | 33.88 | 1.046 |
| Models | DMCG-Frag | | | | DMCG | | | |
| Metric | **COV-R** | **AMR-R** | **COV-P** | **AMR-P** | **COV-R** | **AMR-R** | **COV-P** | **AMR-P** |
| 1000 | 0.451 | 1.488 | 0.101 | 1.493 | 0.412 | 1.540 | 0.073 | 1.501 |
| 5000 | 0.467 | 1.477 | 1.135 | 1.300 | 0.578 | 1.491 | 0.931 | 1.359 |
| 10000 | 7.568 | 1.279 | 8.047 | 1.187 | 6.405 | 1.295 | 7.162 | 1.190 |

$\delta = 0.5$ Å. We acknowledge that DMCG demonstrates excellent performance on this dataset, which consists of smaller molecules. DMCG's direct coordinate regression approach appears particularly effective for QM9's small molecules, where the conformational space is less complex and more amenable to end-to-end prediction. However, it's worth noting that while DMCG excels in COV-R metrics, our FRAGDIFF-T achieves better AMR scores and superior COV-P and AMR-P metrics, indicating higher quality of generated conformers even for these smaller molecules.

To further evaluate the data efficiency of our approach compared to DMCG, we conducted experiments in limited-data scenarios on GEOM-DRUGS. Table 16 presents the results with varying numbers of training samples. The performance gap becomes even more pronounced in these low-data regimes, with DMCG essentially collapsing (near-zero coverage with 1K or 5K training molecules), while FRAGDIFF methods maintain relatively strong performance. This demonstrates that our fragment-augmented diffusion approach dramatically improves sample efficiency when data is scarce—precisely the setting that plagues many real-world drug discovery datasets. Notably, while DMCG-Frag shows slight improvements over the base DMCG model in the extremely low-data regime, these improvements are marginal compared to the substantial gains achieved by FRAGDIFF models.

The observed performance differences can be attributed to fundamental architectural distinctions between the approaches. DMCG employs direct coordinate regression through iterative GNN blocks to predict 3D coordinates in an end-to-end fashion. While effective for whole-molecule prediction, this direct approach may struggle to leverage partial structure information effectively, as it's designed to learn global molecular geometries rather than building them from local structural components. In contrast, diffusion-based methods might offer better compatibility with fragmentation through their probabilistic, step-wise denoising process

that operates in lower-dimensional spaces (torsional or distance geometry), requiring fewer examples to model conformational distributions compared to direct coordinate regression approaches and improves data efficiency.

# E    Limitations, Future Directions and Broader Impact

While our fragment-based pretraining augmentation approach has demonstrated significant improvements in generating accurate and diverse molecular conformers, there are several limitations that present opportunities for future research. First, when applying fragmentation methods as a general data augmentation technique for data-driven computational models, we may encounter unmanageable data volumes, especially when training with large molecular datasets and setting low fragment size thresholds, as discussed in our appendix on fragmentation statistics. This highlights the need for more data-efficient frameworks. Leveraging prior domain knowledge, such as scaffold networks or molecular graphs (Quinn et al., 2017; Nothias et al., 2020; Kruger et al., 2020), could enhance data efficiency during the fragmentation process, reducing the computational burden while preserving essential chemical information.

From a methodological standpoint, while fragment-based augmentation has delivered impressive results within the torsional diffusion framework, there is room for further improvement to fully capitalize on the benefits of data augmentation. The current framework relies on cutting edges in graph structure algorithms, which introduces limitations—such as difficulty in modeling fully connected supramolecular structures where rotating edges alone cannot capture reasonable conformers. Potential solutions include introducing additional variations in Euclidean space, like incorporating ring-connecting edges from junction trees and allowing non-rigid rotational edges that permit changes in relative atomic distances (Jin et al., 2018). Additionally, integrating bond stretching and angle bending components into conformeral energy modeling could address challenges in representing fully connected structures, effectively combining elements of methods like GeoDiff with Torsional Diffusion (Jing et al., 2022; Xu et al., 2022). By exploring the chemical underpinnings of fragment effectiveness, we can gain deeper insights that enable the development of more effective chemical modeling processes, reduce errors, and enhance data learning efficiency through interdisciplinary collaboration. Moreover, our method has the potential to significantly advance computation-driven approaches by greatly increasing the amount of available data, which is crucial for the success of machine learning models. By utilizing our fragmentation approach to augment data and scaling up model parameters, we open new avenues for designing and training larger computational models in physical chemistry, potentially unlocking novel applications in chemical and materials science (von Lilienfeld et al., 2020; Frey et al., 2023; Sadybekov & Katritch, 2023).

