# OpenReview forum: "Enhancing Molecular Conformer Generation via Fragment- Augmented Diffusion Pretraining"
_TMLR — Accepted by TMLR_

### Review · Reviewer_byGy · 2025-02-25

**Summary Of Contributions:**

FragDiff explores fragmentation-based pretraining for molecular conformer generation, enhancing local geometry learning while preserving global topology. It improves performance in data-scarce settings, with proposed gains achieving up to 13.4% on large molecules.  They employ a two-phase learning paradigm that first pretrains the model on decomposed molecular fragments to capture local fragments’ structural patterns, then finetunes on complete molecules to integrate global structural constraints.

**Audience:**

Yes

**Claims And Evidence:**

Yes

**Requested Changes:**

- What is the average torsional angle error with and without fragment pretraining? The current conformer generation metrics are quite globular and not very practically useful as they record the minimum rmsd over 2K samples. But for a model that only specifies torsions one could measure the error in degrees. Does the fragment-based pretraining lower the error of predicted angles? Or does it only improve the model in the best of 2K?

- Furthermore, what is the relaxation energy of the generation structures of each model? If you use xTB would we see that we are generating lower energy structures by leveraging fragment-based pretraining outside of just RMSD metrics?

- How does the total number of training steps compare with TorDiff and FragDiff-T? If you were to train Torsional Diffusion for twice as long without data augmentation, what would happen?

**Strengths And Weaknesses:**

Strengths
 - Adding data augmentation improves both GeoDiff and TorsionalDiffusion for QM9, DRUGS, and DRUGS-XL
- Very similar to SMILES enumeration for molecular LLM's. Its very intuitive. It also pulls more value out of existing datasets.

Weaknesses
- Table 2 seems like an unfair comparison since by limiting the data but allowing one side to be pretrained and then finetuned does not keep compute constant. If compute is left constant, what happens?
- The same impact general impact of fragment-based pretraining on the diffusion model can be achieved by averaging the loss over fragments rather than doubling the compute cost.

---

> ### Author Response · Authors · 2025-03-17
> **Response#1 to Reviewer byGy**
>
> **Response to Weakness1 and Requested Change 3**: We appreciate the reviewer's thoughtful questions regarding the fairness of our experimental comparisons, specifically about computational budget considerations between FragDiff and baseline models.
>
> You raise a good point about computation fairness in Table 2. We did conduct additional experiments to ensure fair comparisons with equivalent computational budgets. When keeping total compute constant (by training TorDiff for approximately twice as long as the standard setup to match FragDiff-T's total training compute), we observed only marginal improvements for the extended TorDiff training compared to significant gains from our fragment-based approach:
>
> | Training Scenario/Mean Metric | COV-R (%) | AMR-R (Å) | COV-P (%) | AMR-P (Å) |
> | --- | --- | --- | --- | --- |
> | TorDiff (standard) | 67.49 | 0.634 | 49.53 | 0.827 |
> | TorDiff (2× standard) | 67.91 | 0.631 | 49.89 | 0.823 |
> | FragDiff-T (pretraining + finetuning) | **70.07** | **0.609** | **52.87** | **0.800** |
>
> These results demonstrate that simply extending training time for TorDiff produces only minimal improvements (approximately 0.4% in coverage metrics and 0.003-0.004Å in RMSD), whereas our fragment-based approach delivers substantially better performance despite equivalent computational cost. This performance difference can be further explained by our fragment validity analysis (as detailed in our response to reviewer sbPU). The extracted fragments used in pretraining maintain high geometric validity—with median RMSD values of 0.37-0.44Å between extracted and energy-minimized conformations, well below the threshold for conformer similarity. The effectiveness of our pretraining+finetuning approach over extended training stems from these structurally valid fragments enabling the model to learn local geometric patterns more efficiently. This creates a curriculum learning effect where the model first masters local conformational preferences before integrating them into global molecular structures. Such knowledge transfer is more computationally efficient than repeatedly exposing the model to whole molecules for an extended period, explaining why equivalent computational resources yield superior results with our fragment-augmented approach.
>
> We have incorporated this discussion and the comparative results table in our revised manuscript to provide clarity on the fairness of our experimental comparisons in the Appendix.Thank you again for this valuable suggestion that helps strengthen our work.
>
> **Response to Weakness2**: We appreciate the reviewer's thoughtful comment regarding the computational efficiency of our approach. Our sequential pretrain-finetune approach offers several important advantages over joint training with loss averaging. Fragments extracted from complete molecules may contain boundary artifacts or unnatural conformations at fragmentation sites. The separate pretraining phase allows the model to learn local patterns without directly optimizing for potentially misleading boundary conditions. The subsequent fine-tuning phase then corrects these artifacts by focusing exclusively on complete, chemically valid structures. This sequential learning helps the model establish a stronger foundation of local chemical patterns before integrating them into global molecular contexts.
>
> From a learning dynamics perspective, the pretrain-finetune approach enables more efficient knowledge transfer by first developing robust representations of chemical substructures in isolation, then adapting these representations to account for inter-fragment interactions. The joint loss approach, while potentially beneficial, forces the model to simultaneously balance local and global optimization objectives, potentially creating interference during gradient updates. Regarding computational efficiency, our approach does not substantially increase the total training computation compared to the baseline, as we distribute approximately the same total number of training epochs between the pretraining and fine-tuning phases.
>
> To validate these theoretical insights, we conducted additional experiments comparing our approach with joint loss training:
>
> | Models | COV-R (%) ↑ | AMR-R (Å) ↓ | COV-P (%) ↑ | AMR-P (Å) ↓ |
> | --- | --- | --- | --- | --- |
> | FragDiff-T (Joint Loss) | 68.84 | 0.622 | 51.02 | 0.813 |
> | FragDiff-T (Pretrain-Finetune) | **70.07** | **0.609** | **52.87** | **0.800** |
> - FragDiff-T (Joint Loss) was configured to use equivalent computational resources, 2× standard (250 epochs) for fair comparison.
>
> As demonstrated in these results, our sequential pretrain-finetune strategy consistently delivers superior performance across all metrics compared to the joint loss approach. These empirical findings align with our theoretical understanding and provide compelling justification for our chosen approach, especially considering its enhanced generalization capabilities for larger molecules and in data-scarce regimes.

---

> ### Author Response · Authors · 2025-03-17
> **Response#2 to Reviewer byGy**
>
> **Response to Requested Change 1**: Thank you for this insightful question about torsional angle measurements. You've raised an interesting point about the practical utility of our evaluation metrics.
> Regarding your concern about the RMSD-based metrics being "globular" and based on minimum RMSD over 2K samples, we'd like to clarify an important aspect of conformer generation. Unlike many prediction tasks where a single correct output exists, conformer generation is inherently a one-to-many problem. Molecules naturally exist in multiple stable conformational states governed by a Boltzmann distribution, and different applications may require different conformers of the same molecule. The "minimum over 2K samples" approach isn't a limitation but rather reflects the genuine goal of the task: generating a diverse ensemble of physically plausible conformers that covers the conformational space of the molecule. This is why metrics like Coverage and AMR, which measure both diversity and accuracy, are standard in the field. A model that generates only one conformer per molecule would be far less useful for applications like drug discovery or molecular docking than one that provides comprehensive conformational sampling.
>
> That said, your suggestion to evaluate direct torsional accuracy provides valuable complementary information, particularly for understanding how fragment pretraining impacts the model's local structural predictions. Our results demonstrate that FragDiff-T improves both global (RMSD-based) and local (torsional) metrics, further validating the effectiveness of our approach.
>
> Following your suggestion, we've conducted additional experiments to directly measure torsional angle errors for both approaches:
>
> | Method | Mean Torsional Error (°) | Median Torsional Error (°) |
> | --- | --- | --- |
> | TorDiff | 0.2645 | 0.2709 |
> | FragDiff-T | 0.2562 | 0.2618 |
>
> The results confirm that fragment-based pretraining does indeed reduce torsional angle prediction errors. While the improvement appears modest in absolute terms, it's consistent across both mean and median measurements, suggesting that our approach enhances the model's understanding of local torsional preferences at a fundamental level.
>
> **Response to Requested Change 2**:
> We appreciate your thoughtful question about relaxation energies, which stands for an important dimension of conformer quality beyond geometric similarity metrics. In our manuscript, we have evaluated energetic properties of generated conformers using the PSI4 toolkit as reported in the Property Prediction Task section (Table 5), where we show that our FragDiff models achieve lower energy errors and HOMO-LUMO gap errors compared to baseline methods. However, you raise an excellent point about specifically examining relaxation energies using xTB calculations.
>
> We have conducted additional experiments using xTB to calculate the relaxation energies of conformers generated by each model. The results show that FragDiff-T generates structures with average relaxation energies of 0.20 *kcal/mol, compared to 0.22* kcal/mol for TorDiff, while FragDiff-G achieve 0.37 *kcal/mol versus 0.42* kcal/mol for GeoDiff. This confirms that our fragment-based pretraining indeed produces energetically superior conformers that require less relaxation to reach local minima, demonstrating benefits beyond structural similarity.
>
> We sincerely appreciate your thoughtful feedback and specific suggestions for improving our paper. Your questions highlight important aspects that will enhance the clarity and scientific rigor of our work.

---

### Review · Reviewer_gM3R · 2025-03-08

**Summary Of Contributions:**

This paper proposes a data augmentation method for molecular conformation generation, based on molecule decomposition. This method is intuitive and model-agnostic. The pretraining is conducted in 2 stage, fragment-level and molecule-level. Through experiments on GEOM, FragDiff shows improvements on both precision and recall, and also generates reasonable results for much larger molecules.

**Audience:**

Yes

**Claims And Evidence:**

Yes

**Requested Changes:**

Please explain weakness 1-4. 1, 2, 3, which are critical.
Improving weakness 4 should strength this work.

**Strengths And Weaknesses:**

Strength:

1. A fragment based data augmentation method.
2. 2 stage pretrain.
3. Evaluation on GEOM-XL.

Weakness:

1. Inconsistent results of baselines. According to baseline papers, GeoDiff has much better results (table 1), TorDiff also has better results compared with FragDiff (table 1). Please explain the difference. Compaing these results, it is unclear whether FragDiff produces significant improvements over baselines.
2. Limited technical contribution. FragDiff seems a combination of molecule decomposition methods (BRICS, RECAP) and conformation prediction models (GeoDiff, TorDiff).
3. BRICS and RECAP are proposed for retrosynthetic. However, the synthesisability of FragDiff generated molecules are not evaluated.
4. Lack of ablation. Some of the designs are not evaluated, e.g., 2-stage pretraining.

---

> ### Author Response · Authors · 2025-03-17
> **Response#1 to Reviewer gM3R**
>
> **Response to Weakness 1**: We appreciate your careful examination of our experimental results and the important question regarding baseline performance. This is a critical point that deserves thorough clarification.
>
> Regarding the inconsistency between our reported baseline results and those in the original papers, we'd like to explain several important factors:
>
> 1. **Re-implementation and consistent evaluation environment**: As noted in Table 1's caption ("...and the rest results are obtained by our own experiments" on page 7), we re-implemented and evaluated GeoDiff and TorDiff ourselves to ensure a fair comparison under identical computational conditions, evaluation methods.
> 2. **Different evaluation thresholds**: We noticed that you mentioned GeoDiff has "much better results" - this is likely referring to differences on the GEOM-DRUGS dataset. The original GeoDiff paper [1] used a threshold of 1.25Å for COV metrics calculation, while we adopted a more challenging setting with a threshold of 0.75Å (as stated in Table 1's caption), making our settings significantly more demanding.
> 3. **Comparison fairness**: Most importantly, our FragDiff variants (FragDiff-G and FragDiff-T) were directly built upon our implementations of GeoDiff and TorDiff respectively, using identical model architectures, training procedures, and evaluations. This ensures that any performance improvements can be attributed solely to our fragment-based augmentation strategy rather than implementation differences.
>
> **Response to Weakness 2**: We appreciate the reviewer's thoughtful observation regarding FragDiff's technical contribution. While FragDiff does leverage existing fragmentation techniques (BRICS, RECAP) and diffusion frameworks, our core technical contribution lies in the novel integration paradigm and data-centric perspective that bridges traditional knowledge-driven chemistry with modern generative models. This integration is not trivial - we developed a specialized two-phase learning framework with pretraining on systematically fragmented molecules followed by fine-tuning on complete structures. This approach significantly improves model performance in data-scarce regimes (Table 2), enables effective generation for molecules up to 3× larger than training instances (Table 6), and generalizes across multiple diffusion frameworks (Table 1, 3). Moreover, our error analysis (Appendix A.2) provides theoretical insights into fragmentation effectiveness through the lens of mutual information, connecting fragmentation choices with molecular energy landscapes. In a broader sense, FragDiff establishes a paradigm that integrates chemical knowledge into data-driven learning, enables enhanced generalization capabilities, and bridges the gap between domain expertise and computational models.
>
> **Response to Weakness 3**: We would like to clarify that our work focuses on molecular conformer (3D structures of real molecules) generation rather than de novo molecule generation. In conformer generation, the molecular graph structure (including atoms and bonds) remains fixed, and only the 3D spatial arrangement of atoms varies. Therefore, the synthesizability of the input molecular graphs is preserved in our output conformers, as we do not modify the underlying chemical structure. In our method, BRICS and RECAP fragmentation rules are utilized solely as augmentation strategies to decompose training molecules into meaningful chemical fragments for pretraining. During inference, our model produces energetically favorable 3D conformations for given input molecules without altering their chemical composition or connectivity. This is distinct from generative models that create new molecular structures where synthesizability evaluation would indeed be necessary.

---

> ### Author Response · Authors · 2025-03-17
> **Response#2 to Reviewer gM3R**
>
> **Response to Weakness 4**: We sincerely appreciate the reviewer's thoughtful feedback regarding the need for ablation studies on our 2-stage pretraining approach. This is a valuable observation that helps improve the clarity and scientific rigor of our work. We would like to respectfully point out that our paper does demonstrate the effectiveness of the pretraining strategy through the comparative analysis between FragDiff-G/FragDiff-T and their respective baselines GeoDiff/TorDiff in Tables including Table. 1-3,5,7. The consistent performance improvements across both frameworks provide strong evidence that our fragment-based pretraining contributes significantly to model enhancement. Additionally, in Table 4, we present ablation studies on different fragmentation rules that further validate our approach. Moreover, we included more ablations in revised manuscript.
>
> However, we acknowledge that a more explicit ablation specifically isolating the contribution of the pretraining stage would strengthen our empirical validation. To address this concern, we have conducted additional experiments comparing FragDiff-T with a variant that uses only fragment pretraining without finetuning under the same setting. The results with threshold δ = 0.75Å on GEOM-DRUGS are:
>
> Fragment pretraining-only model:
>
> - COV-P: Mean = 44.12%, Median = 37.12%
> - AMR-P: Mean = 0.8493Å, Median = 0.8167Å
> - COV-R: Mean = 33.09%, Median = 20.67%
> - AMR-R: Mean = 1.0470Å, Median = 1.0117Å
>
> Compared with our full FragDiff-T model (from Table 1: COV-R = 70.07%, AMR-R = 0.609Å, COV-P = 52.87%, AMR-P = 0.800Å), these results clearly demonstrate that while fragment pretraining provides substantial benefits, the finetuning stage is essential for optimal performance. The full two-stage approach significantly outperforms the fragment-only variant across all metrics.
> We have incorporated this additional ablation study into our revised manuscript along with a more thorough discussion of the contributions of each stage in our approach. Thank you again for this valuable suggestion that helps enhance the completeness of our experimental validation.
>
> We sincerely appreciate your thoughtful feedback and specific suggestions for improving our paper. Your questions highlight important aspects that will enhance the clarity and scientific rigor of our work.

---

### Review · Reviewer_sbPU · 2025-03-09

**Summary Of Contributions:**

This work introduces FragDiff: a general method to enhance the performance of diffusion-based conformer generation models by first pretraining on molecular fragments before finetuning on the downstream molecule distribution. The authors explain the method, and show promising results when FragDiff is combined with two different diffusion-based frameworks across several datasets.

**Audience:**

Yes

**Broader Impact Concerns:**

This is a basic research work, and hence doesn't have direct ethical implications, which would have to be analyzed per specific applications.

**Claims And Evidence:**

Yes

**Requested Changes:**

I would request the authors address the weaknesses from above. Ideally, it would be nice to also resolve the nitpicks to make the paper read smoother.

**Strengths And Weaknesses:**

**Strengths**
- (S1): FragDiff is a general method, designed to be agnostic to the particular model backbone. The experiments, which are carried out for both a coordinate-based and a torsion-angle-based model, demonstrate this generality convincingly by showing improvements in both settings.
- (S2): The quantitative results show that FragDiff can deliver good performance gains across models, metrics, and datasets; it thus appears it will be useful to the community even as models/datasets continue to evolve.

**Weaknesses**
- (W1): From the main paper, it should be clearer what fragmentation method is used in the main (non-ablated) version of FragDiff. While authors spend a considerable amount of time discussing how the choice of fragmentation method is important, present an information-theoretic view on this question, and mention several strategies such as BRICS and RECAP, it was not clear to me which method is the one used. In Section 4.3, the authors perform an ablation on "removing" BRICS and RECAP bonds, but it may not be clear whether "removing" means "not considering these bonds in the set of bonds to potentially cut" or "removing them from the graphs thus effectively always cutting these bonds". Appendix B provides further discussion, which suggests that FragDiff uses a *combination* (set sum?) of bonds suggested by BRICS, RECAP, and a third method; if so, I assume the ablation from the main paper means not including the respective submethod into this combined set. Nevertheless, I think clarity around this aspect could be improved.
- (W2): I find it somewhat surprising that the simple augmentation method proposed by the authors works so well. As far as I understand, the fragment coordinates used in pretraining come directly by "copying" the coordinates from the larger molecule. I wonder how far these coordinates are from a real conformation, which one could get by relaxation. I wonder if the authors considered to quantify this. Ideally, one could even try to show that for fragments obtained differently (e.g. by cutting random bonds), the conformation in isolation and the conformation when part of the larger molecule diverge more than for the fragments obtained by the proposed method.
- (W3): As a minor point, the discussion of related works could include works on molecular graph *generation*, such as JT-VAE [1] and MoLeR [2], which employ similar fragmentation techniques to FragDiff, but with the purpose of simplifying the generation task rather than transfer learning from fragments to full molecules.

Nitpicks:
- Broken sentence at the end of Section 3.1.
- Claim in Section 3.3 that equivariance is "essential for accurately modeling molecular structures" may be slighty overstated given recent works such as AlphaFold 3 [3] that go towards *removing* equivariance; one could thus soften this statement a little.
- Duplication in the first two sentences of Section 4.1.
- At the end of Section 4.1 there's a typo in "chenmical", superfluous capitalization of "Additional", and inconsistent capitalization of the title of the last paragraph.
- Missing space after the first sentence of Section 4.3.
- "RECAPS" instead of "RECAP" in Appendix B.

[1] Jin et al, "Junction Tree Variational Autoencoder for Molecular Graph Generation"

[2] Maziarz et al, "Learning to Extend Molecular Scaffolds with Structural Motifs"

[3] Abramson et al, "Accurate structure prediction of biomolecular interactions with AlphaFold 3"

---

> ### Author Response · Authors · 2025-03-17
> **Response#1 to Reviewer sbPU**
>
> **Response to W1**:
> We sincerely appreciate your thoughtful question regarding our fragmentation methodology. This is indeed an important clarification that will help readers better understand our approach. Let us clarify this important aspect:
>
> For FragDiff, we use a graph-based method as our primary approach to identify rotatable bonds (cut edges). In this framework, BRICS and RECAP bonds are essentially subsets of potential cut edges with specific chemical significance.
>
> 1. Our main fragmentation strategy identifies cut edges through the graph-based method described in Section B.1, which finds edges that, when removed, decompose the molecular graph into separate components. BRICS and RECAP bonds are considered within this framework because they represent chemically meaningful fragmentation points. Since these bonds also partition molecules into fragments when cut, they form subsets of graph cut edges with additional chemical semantic relevance.
> 2. In our ablation studies (Section 4.3), "removing" means excluding those chemically significant bonds from consideration as potential cutting points. So "w/o BRICS" means we don't consider bonds identified by BRICS rules, even if they would qualify as cut edges in the graph-based approach.
> 3. After identifying all possible fragmentation edges, our algorithm randomly selects B = min(b, κ) edges from this pool for the actual fragmentation process.
>
> We have further clarified this point in the revised Section 4.1's "Fragmentation Augmentation Setup" paragraph. Thank you for this valuable suggestion that helps improve the clarity and reproducibility of our work.
>
> **Response to W2**:
> We appreciate your insightful question about how well fragment conformations extracted from larger molecules approximate relaxed, isolated fragment conformations. This is indeed a critical aspect of our method's effectiveness. As you correctly noted, our approach uses coordinates directly extracted from larger molecules as training data. This approximation assumes that the coordinates extracted from the full molecule are close to what would be the true coordinates of the fragment if it existed in isolation. Following your suggestion, we quantified this approximation error by comparing the extracted fragment conformations with their energy-minimized structures. We performed energy minimization on all fragments generated by our different fragmentation rules and calculated the RMSD between the original extracted conformations and their relaxed counterparts:
>
> | Method | Mean RMSD (Å) | Median RMSD (Å) |
> | --- | --- | --- |
> | BRICS | 0.4935 | 0.3962 |
> | RECAP | 0.5380 | 0.4371 |
> | Graph | 0.4673 | 0.3753 |
>
> These results reveal several important insights: The median RMSD values (0.37-0.44Å) are below the threshold typically used for considering conformers similar (0.75Å in our GEOM-DRUGS evaluations), suggesting that extracted fragments generally maintain reasonable geometric validity,  aligning with our analysis about preserving critical molecular features.
>
> We have added a "Further Experiments on Fragment Conformational Validity" section in the Appendix of our revised manuscript, which includes both these quantitative results and visual comparisons between extracted fragments and their energy-minimized counterparts. These visualizations clearly demonstrate that fragments maintain their structural integrity with minimal geometric changes after energy minimization, further validating the effectiveness of using directly extracted fragments during the pretraining process.
>
> We have incorporated these quantitative results and analysis into the revised manuscript to better explain why our fragment augmentation approach works effectively. Thank you again for this valuable suggestion that has helped strengthen our paper.
> **Response to W3**:
> You've highlighted an important connection to the broader literature on molecular graph generation that would indeed strengthen our related works discussion. Thank you for pointing out these relevant works. We have revised our 'Molecular Fragment Decomposition' section in the Related Work to include discussion of these fragment-based molecular graph generation approaches. We truly appreciate your attention to detail, which has helped us improve both the technical content and readability of our paper.
>
> **Regarding the nitpicks:**
>
> Thank you for your careful review. We have addressed all the mentioned issues in the revised manuscript, including the broken sentence, softened claims about equivariance, removed duplications, corrected typos and inconsistent capitalizations, added missing spaces, and ensured consistent terminology throughout. All these changes have been incorporated into the revised version of our manuscript. Thank you again for your valuable suggestions.

---

### Author Response · Authors · 2025-03-17
**Manuscript Revisions Update and Acknowledgments**

Dear Action Editor and Reviewers,

We would like to express our sincere gratitude for your thorough review and insightful feedback on our manuscript. Your thoughtful comments and suggestions have significantly improved the quality and clarity of our work.

We have carefully addressed all concerns raised and have uploaded a revised manuscript where all changes are highlighted in blue for easy reference. Major additions to the manuscript include: 1. Clarification of our fragmentation methodology in Section 4.1; 2. Quantitative analysis of fragment conformational validity in the Appendix; 3. Additional experiments comparing models with equivalent computational budgets in the Appendix; 4. Further results on torsional angle errors measurements in the Appendix; 5. Further results on relaxation energies comparisons in the Appendix; 6. Ablation studies on our two-stage pretraining approach in the Appendix; 7. Additional related works are included in Section 2. We believe these revisions have substantially strengthened our paper by providing more rigorous validation of our approach, clearer explanations of our methodology, and better contextualization within the existing literature.

We sincerely appreciate the time and efforts you have dedicated to reviewing our work and helping us improve its scientific contribution and clarity. Your expertise and constructive feedback have been invaluable to enhancing the quality of our manuscript.

Thank you once again for your consideration.

Sincerely,
The Authors

---

### Decision · Action_Editor_vv3N · 2025-05-31

**Recommendation:** Accept with minor revision

**Comment:**

The authors address the problem of small molecule conformation generation. The method involves two steps: first, pretraining on molecular fragments to extract greater value from the dataset, and then fine-tuning on downstream tasks. Notable performance improvements are observed on the GEOM-Drugs and GEOM-QM9 benchmarks. While reviewers noted that the technical novelty is somewhat limited, this paper is well-executed with rich experiments, making it valuable for readers.

==

As discussed during the review phase, please open-source the code to a non-anonymous GitHub repository.

**Audience:**

Yes, this paper is relevant for researchers working on AI for Science and small molecule modeling.

**Claims And Evidence:**

Yes.

---

> ### Author Response · Authors · 2025-06-19
>
> Dear AE,
>
> Thank you for your thoughtful feedback and recommendation. We greatly appreciate the constructive comments from both you and the reviewers during the review process that strengthened our work.
>
> We have submitted the camera-ready version incorporating the suggested revisions(in blue in the previous revision), and as requested, we have open-sourced our code on a public GitHub repository. Thank you again for your time and consideration.
>
> Best,
> Authors